# Desertification, Resilience and Re-greening in the African Sahel – A matter of the observation period?

**Hannelore Kusserow**

Department of Earth Sciences, Institute of Geographical Sciences, Remote Sensing and Geoinformatics, Freie Universitaet Berlin

*Correspondence to*: H. Kusserow (lolotondi@geog.fu-berlin.de)

**Abstract.** Since the turn of the millennium various scientific publications have been discussing a re-greening of the Sahel after the 1980's drought mainly based on coarse-resolution satellite data. However, own field studies
suggest that the situation is far more complex and both paradigms, the "Encroaching Sahara" and the "Re-greening Sahel", need to be questioned.

The article discusses the concepts of desertification, resilience and re-greening by addressing four main aspects: (i) the relevance of edaphic factors for a vegetation re-greening, (ii) importance of the selected observation period in the debate of Sahel greening or browning, and (iii) modifications in vegetation pattern as
possible indicators for ecosystem changes (shift from originally diffuse to contracted vegetation patterns).

The data referred to in this article cover a time period of more than 150 years and include the author's own research results from the early 1980s until today. A special emphasis, apart from field work data and remote sensing data, is laid on the historical documents.

The key findings summarised at the end show the following: i) vegetation recovery predominantly depends on
soil types; ii) when discussing Sahel greening vs. Sahel browning, the majority of research articles only focuses on post-drought conditions. Taking pre-drought conditions (before 1980s) into account, however, is essential to fully understand the situation. Then botanical investigations and remote sensing based time series clearly show a substantial decline in woody species diversity and cover density compared to pre-drought conditions; iii) self-organised patchiness of vegetation is considered to be an important indicator for ecosystem changes.

**Keywords**

Desertification, Re-greening, Resilience, Self-organised patchiness, Ecosystem engineering.

## 1 Introduction

The long-lasting and sometimes heated scientific debates on Sahel greening vs. Sahel browning inspired the author to critically analyse the existing research results from the viewpoint of over 30 years of own research activities in the African Sahel. The article discusses four key indicators relevant for the greening-browning
discussion: (i) the relevance of edaphic factors for a vegetation re-greening, (ii) plant species changes as a function of the selected observation period, (iii) remote sensing based vegetation changes as a function of the selected observation period, and (iv) modifications in vegetation pattern as possible indicators for ecosystem changes (shift from originally diffuse to contracted vegetation patterns). The discussion is based on own research results and contributions that are put into context of the scholarly debate of the issue. Thus, each section starts
with a description of own contributions followed by an analysis of results of other authors. The article is structured as follows:

Besides an introduction into the motives of the author and the overall structure of the article, the current Sect. 1 provides a brief overview of the African Sahel. Sect. 2 gives a short introduction into data and research methods developed and applied by the author as well as an overview of data and methods of other researchers, cited in
this article.and research methods developed and applied by the author. Sect. 32 provides a general overview of the history of the desertification debate and concepts of desertification, resilience and re-greening. Sect. 4 discusses key factors for the greening/browning discussion as follow: Sect. 4.1 emphasises the relevance of edaphic factors for a vegetation re-greening with examples from Mali, Burkina Faso and Niger. Sect. 3 discusses

the relevance of edaphic factors for a vegetation re-greening with examples from Mali, Burkina Faso and Niger. Sect. 4.2 deals with plant species changes with examples from Mali and Dafur. Sect. 4.35 analysies the concept of a "re-greening Sahel" based on NOAA-AVHRR, GIMMS3g, MODIS, SPOT VGT studies with examples from Mali. Sect. 4.46 discusses changes in vegetation pattern and self-organised patchiness with examples from Mali, Niger, Burkina Faso and Darfur/Sudan. Sect. 57 summarises the findings and provides key messages and new aspects.

The following gives an introductory description of the current and historical situation of the African Sahel and a short summary of own research methods.

The Sahel countries Mauretania, Senegal, Mali, Burkina Faso, Niger, Chad and Sudan are well known for recurring droughts, desertification and lowest human development indices (UNDP, 2015). The region is also characterised by largest population growth rates worldwide (DSW, 20176; UNICEF, 2014) and is affected by growing numbers of jihadist extremists and illicit activities, including arms, drugs and human trafficking, estimated to generate $3.8 billion annually (ICG, 2015). Sahelian population rely to more than 85% on a pure subsistence economy (agriculture and pasture; Krings, 1991a, 2006). As one part of the natural resources base, the ligneous vegetation cover is the fundamental source for energy supply, construction, forages and medicine. More than 90% of the entire Sahelian population which now amounts for approx. 12017 million people, (DSW, 20176) depend on wood and charcoal (Krings, 2006; IEA, 2014; Mortimore, 2016). Monitoring of its spatial distribution dynamics can provide useful information for decision makers and early warning systems.

A selected observation period of thirty years (post-drought) may lead to an evaluation of the Sahelian woody cover that is different from an observation period of 100 years which includes pre-drought conditions. When evaluating Sahel greening vs. browning, the use of Earth observation (EO) tools is restricted to approx. 35 years (NOAA-AVHRR) or less (12 years, MODIS). Landsat archive offers an observation period of now 44 years, aerial photographs provide a historical view of approx. 65 years. The archive of meteorological data started in Senegal in the 1880s, in Niger around 1900 and in other Sahelian countries in the early 1910s or later. Botanical data have been available since 1900. Information on the state of the ecosystem prior to 1900 can be extracted from reports written by European travellers.

The focus of this paper is on three main aspects: (i) How can natural resources, specifically wood resources, be assessed by using documents of different sources of more than 150 years? (ii) What conclusions can be drawn with respect to the still ongoing "greening" vs. "browning" discussion? (iii) Are there indicators for an ecosystem change?

The eco-climatic borders of the Sahel can be defined as the 100 ±50 mm isohyets in the North and 600 mm in the South. The Sahel stretches across Africa over 400 to 600 km wide and nearly 6000 km long covering an area of approximately 3 million km² (Le Houérou, 1989). There are two major mechanisms defining the Sahel zone. The amount of rainfall is one criterion and this comprises a North/South shift according to a surplus or deficit of precipitation. Sahelian rainfall is notoriously unreliable and is characterised by strong inter-annual variability (Lebel and Ali, 2009). Mainguet (1999) documented the shifting of isohyets and discussed amplitudes of displacements of isohyets up to 400 km to the South during the discontinuous drought of 1968–1985. Lebel and Ali (2009) found a shift of 200 km to the South for the drought period (1970–1989) compared with the preceding wet period (1950–1969).

The other criterion for characterisation of the Sahelian zone is based on vegetation. The key factors are: species composition and vegetation distribution. Le Houérou (1989) listed the distribution of common trees, shrubs and perennial grasses in the various ecoclimatic zones between the Sahara and the Equator. The subzones "Saharo-Sahelian" (100–200 mm), "Sahel zone proper" (200–400 mm) and "Sudano-Sahelian" (400–600 mm) are characterised by specific floristic and vegetation distribution grounds, wildlife and livestock repartition as well as land use patterns.

The Northern Sahel predominantly comprises of extended grasslands with isolated thorny trees and shrubs. The region is occupied by nomadic and transhumant pastoralists (Krings, 2006). The Sahelian subzone is dominated by grassland and bush/tree savannas with drought resistant species (evergreen or semi-evergreen) and shows small-scale sedentary farming and semi-nomad farming systems. Farming is based on subsistence food crops as millet, sorghum, cowpea, groundnuts and horticulture products (e.g. tomatoes, onions, water melon, okra, mango). The southern Sahel has potential for crops and livestock husbandry and are characterised by a mixture of Sahelian and Sudanian species (e.g. Breman and Kessler, 1995; Von Maydell, 1986; Le Houérou, 1989; Schulz and Pommel, 1992).

The 100 mm isohyet as the approximate line between the Sahara and the Sahel roughly corresponds with the borderline between contracted and scattered vegetation, defined by Monod (1954). Contracted vegetation ("mode contracté") indicates that vegetation is concentrated in depressions and water courses. This pattern is characteristic for arid ecosystems. Scattered vegetation or "mode diffuse" refers to a more continuous vegetation cover on different soil types and is representative for savanna systems (Monod, 1954).

Nicholson et al. (2012) provide a semi quantitative precipitation dataset for the nineteenth century adding these data to the more modern gauge data. According to their findings a severe and long lasting drought could be documented for the beginning of 19ths century, followed by a moderate recovery in rainfall mixed with some dry years. The 20ths century is well documented. Four drought periods (1908–1914; in the 1940s, beginning of 1970s and 1980s) and one humid period (1950s) can be distinguished (e.g. Nicholson, 1981, 1989; Reichelt, 1987; Druyan, 1989; Mainguet, 1991; Nicholson et al., 1998).

Following the drought period of the early 1980s, a slight recovery in rainfall has been observed (Nicholson, 2005). This recovery is limited and still lower than the 1950–1989 average (Kusserow and Oestreich, 1998). Changes in the characteristics of the rainfall regime have additionally been observed. There is less spatial coherence and less temporal persistence. The contrast between a dryer western Sahel and a wetter eastern Sahel is becoming more significant (Lebel and Ali, 2009; Nicholson, 2013). Sanogo et al. (2015) found a statistically significant positive rainfall trend between 1980 and 2010, however, not reaching the degree of wetness of the 1950s.

During late Quaternary the Sahara advanced to the South several times showing the largest expansion during late Pleistocene some 16,000/17,000 years ago (Ogolian desert) and retreated again (Reichelt et al., 1992). For the last millennium the authors found a southward shift of isohyets by 25–30 km per century.

The human impact in the Sahel region started about 7000 years ago (Schulz and Pommel, 1992). The authors discuss an anthropogenic formation of the Sahel from 4000 BP on as a result of cattle keeping, and small holdings with traditional agrarian systems. These small-scale farming consisted of exploitations of fruit trees and field crops in park systems as well as energy supply and metal production using wood resources. Principal instrument for clearing was and still is fire. The transformation of the landscape resulted in a creation of a savanna system like the present Sahel, evolved from the Holocene transition of Sudanian to Saharian vegetation. Large parts of the western Sahel countries have formed part of big empires since around 800 B.C. (Krings, 1982, 2006; Ki Zerbo, 1992; Devisse and Vernet, 1993; Kusserow, 1994, 1995; Hofbauer, 2013). European travellers like Mungo Park and Oskar Lenz (Hoffmann, 1799; Lenz, 1892) reported cultivation of maize in the area of today's Canal du Sahel where nowadays even the cultivation of millet is problematic (Kusserow, 1995). The references of a more humid period had changed in the second half of the last century. Since the 1970s drought period the Sahel stands for desertification and increasing poverty (UNCOD, 1977).

To assess the vegetation dynamics in the Sahel, the author developed the following research methods. A satellite-based woody vegetation interpretation key for semi arid Mali was established in 1985/1987 (enhanced version in 1992; Kusserow, 1986, 1994). The interpretation key consists of structural and floristic criteria of the ligneous vegetation cover combined with morpho pedological characteristics. Applying visual interpretation techniques the method allows the mapping of Sahelian woody vegetation as well as a distinction of fields and fallow land. Dry season Landsat data were used to better discriminate woody from herbaceous vegetation cover. GIS techniques and on screen digitizing (available since the end of the 1990s) were applied for research projects in Niger, Mauretania, Chad and Darfur/Sudan (Kusserow, 2001, 2002, a, b, 2005, 2014). Based on these techniques, changes in vegetation pattern and density were quantified and mapped in form of change detection maps, showing "winners" (predominantly agriculture) and "losers" (vegetation) as basis for planners and decision makers. The method was established in 1999 and has been further developed from 2001 on (Kusserow, 2001; Kirsch-Jung and Kusserow, 2002; Kusserow, 2014).

Two different Landsat satellite data sets were referred to in this article. The first data set includes Landsat MSS, TM, ETM+ and OLI data and was ordered for free from the United States Geological Survey through Earth Explorer (http://earthexplorer.usgs.gov/). These data are already pre-processed and systematically corrected. The data sets comprise:

- Landsat MSS (example Burkina Faso and Niger: year 1973; RGB = 4-2-1; Sect.3)
- Landsat 4-5 Thematic Mapper (TM, example Niger: year 1984, 2002; RGB =4-3-2; Sect. 6).
- Landsat (ETM+, example Niger: year 2009; RGB =4-3-2; Sect. 6).
- OLI (example Niger: year 2013, 2014, example Burkina F.: year 2013; RGB =5-4-3; Sect. 6).

## 2  Data and Methods

The author's investigations are based on: (i) EO tools (aerial photographs and high resolution Landsat/SPOT/IRS satellite data). (ii) Botanical in situ measurements and (iii) Extended ground truth since 1985.

(i) EO tools: To assess the vegetation dynamics in the Sahel, the author developed the following research methods. A satellite based woody vegetation interpretation key for semi-arid Mali was established in 1985/1987 (enhanced version in 1992; Kusserow, 1986, 1994). The interpretation key consists of structural and floristic criteria of the ligneous vegetation cover combined with morpho-pedological characteristics. Applying visual interpretation techniques the method allows the mapping of Sahelian woody vegetation as well as a distinction of fields and fallow land. Dry season Landsat data were used to better discriminate woody from herbaceous vegetation cover. GIS techniques and on-screen digitizing (available since the end of the 1990s) were applied for research projects in Niger, Mauretania, Chad and Darfur/Sudan (Kusserow, 2001, 2002, a, b, 2005, 2014). Based on these techniques, changes in vegetation pattern and density were quantified and mapped in form of change detection maps, showing "winners" (predominantly agriculture) and "losers" (vegetation) as basis for planners and decision makers. The method was established in 1999 and has been further developed from 2001 on (Kusserow, 2001; Kirsch-Jung and Kusserow, 2002; Kusserow, 2010; Kusserow, 2014).

Two different Landsat satellite data sets were referred to in this article. The first data set includes Landsat MSS, TM, ETM+ and OLI data and was ordered for free from the United States Geological Survey through Earth Explorer (http://earthexplorer.usgs.gov/). These data are already pre-processed and systematically corrected. The data sets comprise:

- Landsat MSS (example Burkina Faso and Niger: year 1973; RGB = 4-2-1; Sect. 4.1)
- Landsat 4-5 Thematic Mapper (TM, example Niger: year 1984, 2002; RGB =4-3-2; Sect. 4.4).
- Landsat (ETM+, example Niger: year 2009: RGB =4-3-2; Sect. 4.4).
- OLI (example Niger: year 2013, 2014, example Burkina F.: year 2013; RGB =5-4-3; Sect. 4.4).

The data were displayed with ENVI 4.7 (UTM 31/WGS84) using the default of a 2% linear stretch applied to each image band and for all data. The second one in Sect. 4.3 comprises historical datasets of Landsat MSS and TM (Example Mali, dates: 1976, 1985 and 1991). The raw data were bought from US Geological survey in 1985 and 1991 and were processed in 1991 using ERDAS 7.4.1 and 7.5 (Kusserow, 1990, 1994, 1995). RGB = 4-2-1 (MSS) and 4-3-2 (TM). Processing steps were:

- Correction of six line effect for the MSS data.
- Master scene from 1991 was relatively corrected (haze correction).
- Geometric correction was conducted on the basis of the topographic map (UTM 31/WGS84) by using 17 way points (scene subset: 70 km x 50 km).
- MSS scenes were geometrically corrected on the basis of the master scene (TM).
- Relative calibration of the three datasets (radiometric correction) was performed by look up table modification (calibration on the basis of two test sites, showing no temporal variation).

Note: vegetation classification was done using visual interpretation techniques based on detailed in-situ knowledge.

In addition to Landsat data, change detection assessments were performed, using aerial photographs from the 1950s and 1970s (Kusserow, 1994) and aerial photographs and kite photographs from the 1970s and 1990s (Kusserow and Haenisch, 1999).

(ii) Botanical in situ measurements and further investigations: The author carried out botanical inventories in 1985, 1991 and 1992 in the Canal du Sahel area in Mali. The inventory included measurements of all woody individuals in a 0,1 ha test plots (height, diameter at ground and breast height and crown diameter) as such providing detailed information about vegetation composition and density in the area (Kusserow, 1986, 1994, 1995). Further investigations included multitemporal analyses of soil algae crusts (Hahn and Kusserow, 1998) and molecular genetic studies of wild Sahelian forages (Kusserow et al., 1999) as well as analyses of rainfall data (Kusserow and Oestreich, 1998).

(iii) Extended ground truth since 1985: The term „ground truth" includes detailed field checks of preliminary satellite image interpretation, in particular documentation of landscape elements, monitoring of woody species and species constitution as well as different types of land use. All observations were documented by photos and GPS coordinates (before 1993: classical approach with topographic maps and notation of driven kilometres). Transect observations included detailed notes, photos and GPS coordinates during the field surveys.

Attention will be drawn especially to the so-called "Darfur Project", started in 2010. This still ongoing research project is aimed to prepare a multi-layered dynamic state of the art natural resources and land use database (NRDB) for Darfur/Sudan. Within this international project, funded by the Sudanese government and carried out by the Munich-based Gesellschaft fuer Angewandte Fernerkundung (GAF) AG, six major topics or layers, including geology and hydrology, geomorphology/soil, socio-economics, eco-biology and land cover/land use forms part of the investigation. The author is responsible for the last two layers. Targeted to elaborate a comprehensive report on the ecology and land use in Darfur/Sudan and to prepare a database and maps (1:250,000), both layers shall provide information for planners and decision makers. The projects tasks were:

- Satellite based interpretation of vegetation and land use classes, scale1:250 000 (2010, 2000, 1970s) and analyses of ecological changes.
- Satellite based interpretation of vegetation and land use classes for selected areas, scale1:50 000 (2009-2011).
- Review of documents (1950s until 2015).
- Field survey (measurements, observations, questionnaires).

ESRI's ARC GIS 9.2 software was used for change detection analyses and map production. Satellite data interpretation (on-screen digitizing) was performed using a tailor-made software (Georover) designed by GAF company. For the plant specimen inventory a systematic sampling scheme was developed. Based on soil-vegetation units derived from satellite imageries and the GPS coordinates, sample sites (size of 20 x 20 m for woody vegetation and 1 x 1 m for herbaceous vegetation) were established to quantify the distribution and relative abundance of plant species. Within the sample sites the following parameters were measured/documented: trees/shrubs maximum height, stem diameter, crown diameter and species status (threatened, rare, common etc.). Despite insecurity in some areas, a total of 665 sample plots which accounts to 85% of the originally selected plots could be successfully measured.

Additional information on natural resources conditions was gained through interviews as part of the socio-economic investigations. Four communities in four states were targeted (urban, rural, nomads and the internally displaced persons [IDPs]). In total 102 communities and 2,547 households in Darfur had been interviewed during May to October 2014. Group discussion (12–20 people) was used as method for data collection. The interviewed persons were between 30 and 50 years old. Moreover, wild life investigations along transects and land use surveys were carried out.

Fig. 1 shows the locations of research projects and transects of in situ observations.

**Figure 1.** Position of own research projects and transects of observations.

Contributions of other authors: Various versions of the NOAA-AVHRR (National Oceanic and Atmospheric Administration – Advanced Very High Resolution Radiometer) data have been used to monitor vegetation trends in the Sahel (e.g. Tucker et al. 1991, 1999; Anyamba and Tucker 2005; Herrmann et al., 2005; Dardel et al.,

2014b; Knauer et al., 2014). Global Inventory Modeling and Mapping Studies (GIMMS) with a very coarse spatial resolution (5-8 km, Mbow et al., 2015, Rasmussen et al., 2015) are applied (Sect. 4.1 and 4.3).

New satellite derived imageries such as the new generation GIMMS-3g with a coarse resolution factor of 8 km; MODIS (Moderate Resolution Imaging Spectroradiometer) available since 2000 with a spectral resolution of 250 m and „SPOT-Vegetation (VGT) with a resolution of 5 km and 1km (since 1999) are used more recently (Anyamba et al., 2014; Brandt et al., 2014a; Mbow et al., 2014; Rasmussen et al., 2014). Fensholt et al. (2004, 2015) and Brandt et al. (2014a, d) used datasets from MODIS, Geoland GEOV1 (5 km resolution) and GIMMS3g (8 km resolution) FAPAR (Fraction of absorbed photosynthetically active radiation) to assess local vegetation trends in Senegal and Mali. Horion et al. (2014) explored how dry season $NDVI_{min}$ can be used as proxy indicator for assessing changes in tree cover density.

Recent studies include biophysical variables like FAPAR and LAI (leaf area index), seasonal vegetation dynamics and land surface phenology (Ivits et al., 2013; Brandt et al., 2014a, d; Fensholt et al., 2015; Gessner et al., 2015; Diouf et al., 2015, 2016). Brandt et al. (2016a, b) apply a phenology-driven model for estimating woody canopy cover in the Sahel at 1/0.5 km resolution scale on the basis of MODIS and SPOT-Vegetation FAPAR data.

Comparative analyses of aerial photographs for vegetation assessments were conducted e.g. by Couteron et al. (1997); Rasmussen (1999) and Barbier et al. (2006). Further methods mentioned in this article comprises:

- Vegetation survey and measurements (e.g. Trochain, 1940, Roberty, 1946; Harrison and Jackson, 1958; Miehe, 1988; Hiernaux et al., 2009a; Gonzales, 2001; Gonzales et al., 2004; 2012; Miehe et al., 2010, Sect. 4.2)
- Questionnaire-based surveys among elder people in the Sahel (e.g. Rasmussen et al., 2001; Gonzales et al., 2004, 2012; Ouedraogo et al., 2010; Brandt et al., 2014c; Sambou et al., 2016; Sect. 4.2)
- Sedimentological and micromorphological investigations including high-resolution pollen diagrams (Ibrahim and Schulz, in press, Sect. 4.2)
- Mathematical models of vegetation growth in semi-arid regions (e.g. Thiéry et al., 1995; von Hardenberg et al., 2001; Lejeune et al., 2002; Rietkerk et al., 2004; Dekker et al., 2007; Gilad et al., 2007; Meron, 2012; Sect. 4.4).

## 3~~2~~ Desertification, Resilience and Re-greening – a general overview

The desertification debate started in the 1970s, caused by attracting attention in the scientific community as a result of the severe drought period in the early 1970s. The first international conference on environmental issues was held in Nairobi in 1977 (UNCOD, 1977). The most frequently-used definition of desertification is defined by the UN Convention to combat Desertification (UNCCD, 1994): "Desertification is land degradation in arid, semi-arid and dry sub-humid areas resulting from various factors, including climatic variations and human activities".

Mainguet (1999) stressed, that any concern to define the term "desertification" will end up in ambiguity. The main question - how to discern an irreversible state of land degradation and degraded levels which are partly reversible - remains still open. Also Rasmussen (1999) pointed to a weakness of the definition and use of concepts. Prince (2016) compared global maps of land degradation and desertification and concluded the absence of reliable maps or means of desertification monitoring ("what is degraded?", "where does it occur?", "how severe is the degradation?").

The concept of an "Encroaching Sahara" came up during the early 1920s when first European scientists visited the region (Bovill, 1921; Stebbing, 1935, 1938). They discussed a growing aridification and established the concept of a southward shift of the Sahara towards the savannas in the south. In the late 1930s, a British- French expedition assessed the "Encroaching Sahara" concept in Niger and found that the vegetation cover has recovered and the tree cover in particular was in a very good status (Jones, 1938). Stebbing (1935, 1938) and others interpreted the post-drought situation as a desert encroachment thus misleading future scientists. The first drought period in the past century lasted from 1909–1915 (Nicholson, 2012) but the early researchers neglected, that the data for such a short time span would rather indicate only climatic fluctuations instead of real climatic crisis (Mainguet, 1991).

A new paradigm regarding desertification emerged at the beginning of the 1990s: the "re-greening Sahel" (Helldén, 1991; Thomas and Middleton, 1994; Nicholson et al., 1998; Mainguet, 1999; Herrmann and Hutchinson, 2005; Olsson et al., 2005; Helldén and Tottrup, 2008; Knauer et al., 2014; Behnke and Mortimore,

2016). This new paradigm is predominantly based on studies using coarse satellite remote sensing data ~~(National Oceanic and Atmospheric Administration – Advanced Very High Resolution Radiometer –~~ (NOAA-AVHRR), monitoring the period 1981 until today (Anyamba and Tucker, 2005; Nicholson et al., 2012; Dardel et al., 2014b). Since the millennium, additional data like MODIS and SPOT Vegetation (VGT) NDVI data have been used (e.g. Herrmann and Tappan, 2013; Dardel et al., 2014a; Brandt et al., 2015; Brandt et al. 2016a, b, 2017a; Tong et al., 2017). Reviews are provided by Higginbottom and Symeonakis, (2014), Knauer et al. (2014) and Mbow et al. (2014; 2015). Still today a very contrary discussion regarding re-greening and degradation/browning remains unsolved (Reichelt et al., 1989; Hein and de Ridder, 2006; Mortimore, 2006; Prince et al., 2007; Hein et al., 2011; Dardel et al., 2014b; Mbow, 2015; Mortimore and Behnke, 2016). Recent studies also discuss a "greening" trend vs. a "browning" trend and found - despite a re-greening - significant ground based indicators for an impoverishment of the ligneous vegetation cover, underlining the need for contextual knowledge (e.g. Herrmann and Tappan, 2013; Brandt et al., 2014a, 2015; Dardel et al., 2014b; Mbow et al., 2015; Spiekermann et al., 2015).

Rasmussen et al. (2006) refer to the apparent contradiction between macro-scale analysis of satellite images and micro-scale field studies. In a recently published article Rasmussen et al. (2015) discussed the reasons behind the conflicting evidence and identified inconsistencies of concepts, methodological problems and sampling biases. In particular different temporal scales play an important role.

Actually, a set of indicators and various sources of information are necessary to assess such complex phenomenon as ecosystem fluctuation. Besides remote sensing data (satellite data, aerial photographs) other documents like botanical surveys from the first part of 20ths century, rainfall data, maps, historical documents, reports and questionnaires should be used for assessing land dynamics. Higginbottom and Symeonakis (2014) who reviewed more than 150 article regarding assessments of degradation, called for a "multi-faceted methodology". The longer the observation period is, the more sound will be the information for identification of long-term degradation processes (Miehe et al., 2010).

The question of decreasing or increasing woody cover is fundamental for people's livelihood. Particularly the resource "wood" as main energy supply plays a key role in ensuring the survival of the local people and curbing emigration. Declining wood resources aggravate the already critical situation in the Sahel states (Ouedraogo et al., 2010).

~~3   The relevance of edaphic factors for a vegetation re-greening~~
**4   Desertification, Resilience and Re-greening – key factors**

**4.1   The relevance of edaphic factors for a vegetation re-greening**

The discussion of a system's ability to recover after drought is a key focus in the desertification/re-greening debate. As learned from scientific literature of the early 1920s and 1930s, the vegetation cover in the Sahel-Sudan ecozone has been recovered from the severe drought period in the beginning of the 20th century.

Own contributions: During numerous field studies in Mali, Burkina Faso and Niger in the 1990s, the author documented and investigated the phenomenon of crusted soils in situ and on the basis of satellite-based vegetation analyses (Kusserow, 1995, 2014, Hahn and Kusserow, 1998, Kusserow and Haenisch, 1999). Based on freely available Landsat series, the monitoring of sterile surfaces in areas in Mali, Burkina Faso and Niger could be continued in recent years. Investigations of three sites in Mali, Burkina Faso and Niger, document the relevance of the soil type for vegetation recovery (Fig. 2).

**Figure 2.** Position of research sites in Mali, Burkina Faso and Niger.

The situation in the field is shown in Fig. 3 (a and b), demonstrating different abilities of vegetation recovery as a function of soil type and protection.

**Figure 3a.** Toukounous, Niger (August 1995).        **Figure 3b.** Canal du Sahel, Mali (August 1994).

The left photo (a) taken from the author at the site in Toukounous, Central Niger, show the ability of the system to recover from drought periods on sandy soils and - this should be emphasised - under protection and

controlled grazing. This site forms part of a national cattle breeding station in the 1990s. Outside of the protected area highly grazed dunes are visible. The grass height is less than 5 cm. Brandt et al. (2014c) also refer to the importance of soil properties in the context of ecosystem resilience.

The right picture (b) shows a largely degraded landscape in Mali (Canal du Sahel area). The fertile upper soil

layer had been removed by wind and water activity, resulting in soil crust formations. Dead branches still fix a sandy layer. If branches are collected by local population the small sandy residuals will be also blown out. Resource protecting measures and management are the only alternative for recovery. Brandt et al. (2014a, b) and Spiekermann et al. (2015) present comparable situations in Senegal and other parts of Mali (Bandiagara).

Dune systems of late Quaternary age are one of the major land types in the African Sahel and have high

importance as one of the main agricultural regions in the Sahel zone. Showing a predominant ENE–WSW orientation, they are extensively cultivated and referred to as "Erg Ogolièn" in the western Sahel and "Qoz" in the eastern Sahel (Le Houérou, 1989; d'Herbès and Valentin, 1997). Satellite images from Mauritania, Mali, Niger, Burkina Faso, Chad and Darfur document the key significance of Quaternary dune systems for rural livelihood (Kusserow, 2014).

The Mare d'Oursi site in Burkina Faso, well known as the "Oursi-dune", and first analysed by Toutain and de Wispelaere (1978), is a good example to discuss resilience, re-greening as well as desertification. De Wispelaere (1990) documented an increase of un-vegetated areas on the Palaeolithic dune systems north of the Mare on the basis of aerial photographs from 1955, 1974–1976 and 1981 and interpreted this development as desertification.

Landsat system's archive with high resolution satellite imageries (<1982: 80 m, 1982–2012/13: 30 m, >2013:

30 m/15 m) offers the possibility of change detection analysis. Images from the early 1970s provide information on pre-drought conditions. Woody cover reflects the situation of the Sahelian ecosystem at the beginning of the drought and the end of the 1950/60s more humid phase (Kusserow, 1986, 1995, 2014).

Own comparative analysesi<s>s</s> of two Landsat imageries recorded in 4 July 1973 and 18 September 2013 indicate clear vegetation pattern changes for eastern Burkina Faso (Mare d'Oursi, Fig. 4). A legend supports

identification of individual land units.

**Figure 4.** The Landsat subsets from eastern Burkina Faso cover an area of approx. 70 km x 30 km.

Mare d'Oursi is visible in the western part of the Landsat OLI imagery (colour infrared composition, RGB=543, vegetation is red) which has been recorded on 18 September 2013. The Palaeolithic dune systems stretching from east to west are well visible. In the northern part, severely eroded areas can be clearly identified. These areas present the Precambrian crystalline basement complex consisting of different series of Precambrian rocks (Carte géologique de L'Oudalan, 1:200 000, BRGM, 1970). The basement complex, still covered by

dense/open vegetation in the 1950s (see Fig. 5) and also in the 1970s (Toutain and de Wispelaere, 1978), is now completely denuded. This banding pattern of actual bare areas stretches over approx. 600–700 km until western Niger (example from western Niger in Sect. 4.4<s>6</s>, Fig. 15).

The Landsat imagery recorded on 4 July 1973 (Landsat MSS, RGB=421) shows the impacts of the 1970s drought period. The Palaeolithic dune system has a very limited vegetation cover due to low rainfall. The

40 vegetation distribution on the Precambrian basement (see yellow box<s>ellipsoid and yellow circle</s>) is still in a homogeneous pattern state. The light purple colour is typical for the predominant woody vegetation cover (see also example Niger) during the drought year 1973.

The comparison of both images indicates an apparent inversion. The dune systems in 1973 appear only sparsely covered by vegetation whereas the Precambrian basement shows a more homogeneous vegetation

cover. A contrary situation is visible in 2013. The former tiger bush covered basement now appears with a clearly fragmented vegetation pattern (see yellow box <s>Sect. 6</s>) the eastern and northern parts are already bare. Due to higher rainfall the dunes show a good vegetation cover and extensive fields. These sandy deposits are extensively cultivated with millet. It is worth noting that the <s>fragmented</s> vegetation patches are <s>is</s> located in the valleys (see reference image from 1973). Due to higher water availability, caused by increased run off from

already crusted higher areas and due to an accumulation of fine soil particles, vegetation is increasing in these parts. Own change detection assessments in the Canal du Sahel area in Mali (aerial photos, dated 1953 and high resolution SPOT satellite data of 1992), documented an increase in run off due to losses in woody cover, triggered by clearing for agriculture and fuel wood (Kusserow, 1994).

The results of the satellite image analysis can be backed up with the extensive work of Toutain and De Wispelaere (1978). Maps of the region document a dense to open savanna vegetation cover. Additional information regarding vegetation cover is given by the topographic map Hombori, (Feuille ND-30-NE, IGN Paris 1961; see Fig. 5). There, the dune systems close to Oursi village is still covered by either savanna or grass savannas (prairie). Major parts of Precambrian basement show pattern vegetation (tiger bush). All maps of West Africa printed by IGN in the early 1960s are based on analyses of aerial photographs (flight periods were the early 1950s). It is worth mentioning that the topographic maps document a great extent of tiger bush pattern as well as savanna and grass savanna vegetation distribution even until 16° N. In contrast to the situation in the 1950s–1970s, the current image shows bare ground with marked vegetation patches~~terns~~.

**Figure 5.** Section of the topographic map "Hombori" (IGN, 1961), indicating vegetation types of the early 1950s. 1: Tiger Bush, 2: Savanna, 3: Prairie (grass savanna). The box shows the position of the satellite image section (see Fig. 4).

Krings (1980) described the ecological situation in Oudalan, Burkina Faso on the basis of extended field surveys with special emphasis on the remarkable cultural geographical changes in the contact zone of nomadic and sedentary ethnic groups. He visited the region between October 1976 and March 1977 and reported species like *Adansonia digitata, Pterocarpus lucens, Commiphora africana* and *Guiera senegalensis* in the tiger bush formation, north of Oursi (see Fig. 6). According to a photo taken January 1977, a dense bush formation with a single large *Adansonia digitata* can still be found there at that time. Krings (1980) mentioned also extended vegetation losses in the tree savanna region in northern Oudalan.

**Figure 6.** Vegetation and Landuse in the north-east of Upper Volta and lower Gourma, Mali, status 1976–1977, modified after Krings (1980). The box shows the position of the satellite image section (see Fig. 4).

The preceding investigations clearly show a re-greening on sandy soils (Quaternary dune systems) but a browning on poorly developed soils of the underlying Precambrian basement. This indicates the ecosystem's ability to recover almost exclusively on Quaternary dunes and not on shallow soils. Mensching (1990) presented a map indicating the occurrence of late Quaternary dune systems throughout the Sahel (Fig. 7). The picture is incomplete since dune systems in Eastern Chad and western Sudan (Darfur) seem to be underrepresented. However, this map gives a first overview of areas with potential for vegetation recovery and areas with other soil types possibly indicating less resilience capability.

**Figure 7.** Position of late Quaternary dune systems in the Sahel according to Mensching (1990).

Contributions of other authors: Many authors documented a re-greening trend on sandy soils and discussed the ability of post drought regeneration as a function of soil type and topographic position (Hiernaux et al., 2009a, 2016; Vincke et al., 2010; Brandt et al., 2014a, b, c, 2015; Dardel et al., 2014b; Rasmussen et al., 2014). All study areas cited in this article and reporting a re-greening trend, as Senegal (e.g. Herrmann and Tappan, 2013; Brandt et al. 2014a, b, 2015; Herrmann and Sop, 2016), Mali/Burkina Faso (Hiernaux et al., 2009a, b; Dardel et al., 2014a, b; Rasmussen et al., 2014; Brandt et al., 2016b) and Niger (Hiernaux et al., 2009a, b; Boubacar, 2016) are situated on the sandy soils of the Quaternary dune systems. For the "Oursi-dune" example, Rasmussen (1999) found an increase in vegetation (bushes and herbs) by comparing aerial photos from 1955, 1974, 1981 and 1996. Regarding the aspect of topographic position, Kaptué et al. (2015) confirmed an increasing woody cover at the watershed scale in the majority of their samples. The drier more northerly Sahelian watersheds in Senegal and eastern Mali appear to show stronger reforestation trends than the more mesic region of western Mali and high human population area near Niamey, Niger. Mainguet (1991) has already pointed out that due to increasing run off towards the valleys vegetation re-growth is improved. The importance of different landscape elements for re-greening processes is also mentioned by Vincke et al. (2010) and Rassmussen et al. (2014). The latter two authors found negative NDVI-pixels on the plateaus and slopes and positive pixels in the valleys. Vincke et al. (2010) reported a distinctive regression of woody vegetation in the high relief areas. Own research studies for western Niger (Kusserow and Haenisch, 1999; Kusserow, 2010, 2014) documented severe woody vegetation losses on lateritic plateaus in southwest Niger.

To conclude: recovery of vegetation widely depends on morpho-pedological factors. A recovery on sandy soils have been often documented (e.g. Hiernaux et al., 2009a), whereas on poorly developed soils and crusted soils less or no regeneration can be found (Hiernaux et al., 2009a; Brandt et al., 2014a, b, 2015; Dardel et al., 2014b; Rasmussen et al., 2014). Hiernaux et al. (2016) concluded for the Gourma region in Mali a strong resilience on sandy soils but a collapse and profound mutation of the vegetation on shallow soils.

## 4.2  Plant species change

A different understanding of re-greening becomes evident when the viewing angle is extended to pre-drought condition. Comparing the period 1980–2015 with earlier years, the so called "re-greening" can be quoted as "dramatic decline" in vegetation due to a much higher floristic composition and vegetation density prior to the 1980s. For West Africa, botanical investigations had been conducted by e.g. Chevalier, 1900; Furon, 1929; Trochain, 1940; Roberty, 1946; Aubrèville, 1949; Monod, 1954; Toutain and De Wispelaere, 1978 (see Fig. 8). For East Africa (Sudan) botanical studies from e.g. Andrew, 1950; Harrison and Jackson, 1958; Ramsay, 1958; Lebon, 1965; Wickens, 1976; Ibrahim, 1980; Miehe, 1988 and various Hunting Technical Reports from the 1950s till the mid of 1990s are available. Fig. 8 presents a compilation of selected authors.

**Figure 8.** Compilation of selected botanical assessments in the West- and the East-African Sahel since 1900.

Own contributions: The author carried out botanical inventories in 1985, 1991 and 1992 in the Canal du Sahel area in Mali. The inventory included measurements of all woody individuals in a 0,1 ha test plots (height, diameter at ground and breast height and crown diameter) as such providing detailed information about vegetation composition and density in the area (Kusserow, 1986, 1994, 1995). These data were compared with botanical inventories, realised by Roberty (1946) in the 1940s. Results showed significant changes in species composition. Roberty (1946) still documented mesic (Sudanian) woody species like *Terminalia avicennioides, Bombax costatum, Pterocarpus lucens, Sclerocarya birrea, Sterculia setigera.* Own investigations show a clear shift in the range of species into more robust (*Combretum glutinosum, Guiera senegalensis*) and arid-tolerant ones as *Commiphora africana. C. glutinosum* and *Guiera senegalensis* (family Combretaceae) are typical invaders on fallow land. Still today large areas of the Sudano-Sahelian ecozones are characterised by these two species. Savannas dominated by Combretaceae seem to develop from dense woodland that has been subject to intensive clearing and wood cutting (Trochain, 1940; Aubréville, 1944, 1949, 1950; Le Houérou, 1989).

Figure 9 shows a shift of isohyets comparing Climatological Normals (CLINO) 1931–1960 with 1961–1990. During CLINO 1 the average rainfall varies between 500–600 mm in the research area, whereas CLINO 2 indicates a significant shift to 300–400 mm which represents a typically Sahelian climate. The average rainfall of 500–600 mm fits into the Sudano-Sahelian ecozone, allowing more mesic species to grow. The ligneous fingerprint clearly indicates that more moisture demanding species with Sudanian provenience had been part of the woody population until approx. the early 1980s (Kusserow and Oestreich, 1998). The post drought conditions favour more drought tolerant species, leading to a selective die back of species (e.g. *Pterocarpus lucens;* Kusserow, 1995). This southward trend of annual rainfall crossing the 600 mm annual rainfall threshold for Sudanian flora (Le Houérou, 1989) and has a tipping point like ecological significance (Maranz, 2009; Kusserow, 2014).

**Figure 9.** Location of average isohyets at CLINO 1 (1931–1960) and CLINO 2 (1961–1990), position of research areas is indicated by the red box (modified after Kusserow and Oestreich, 1998).

One driver of the observed decrease in biodiversity is the lower level of precipitation rates which is/was responsible for the species turnover in the ligneous population towards more xeric species. The other driver is human activity (deforestation, clearing for fields). The widespread occurrence of *Combretum glutinosum* and *Guiera senegalensis* which according to own observations form – together with a few other species – the prevailing ligneous cover in the Sudano-Sahelian ecozones across Africa. Both species are first pioneers on fallow land and clear indicators for former agricultural activity. In the Sahelian ecozone they are represented by *Leptadenia pyrotechnica.* The region between Zinder and Goure in eastern Niger are exclusively covered with this highly drought-resistant species. *Leptadenia pyrotechnica* has Saharian affinities (Le Houérou, 1989) and

form almost pure stands in eastern Niger, indicating former agricultural areas. These areas experienced a long lasting human settlement history as part of the big empire Kanem-Borno (Krings, 1982).

A massive abundance of *Leptadenia pyrotechnica* has been observed during own field surveys in 2009 in the area of Tillabéri (Niger), as could not be seen in the mid of the 1990s. The same development was found around of El Fasher, North-Darfur, Sudan during own field surveys in 2014 as part of the Darfur project, executed by GAF-Munich/Germany and financed by the Sudanese government. The accompanying 60 years old forester, grown up in this area, confirmed millet fields and a much more diverse savanna vegetation as well as a rich fauna having been present in the 1960s. *Leptadenia pyrotechnica* as a typical representative of the eco-climatic "Sahara" and "Saharo-Sahelian" zones (Le Houérou 1989) and classified as ecological indicator species (Miehe et al., 2010) points to changing ecosystem conditions. Also Hiernaux et al. (2009a) reported an increase of *Leptadenia pyrotechnica* in the Gourma region in Mali. Rasmussen (1999) as well found an increase of *Leptadenia pyrotechnica* on the denuded part of the Oursi dunes in Burkina Faso. He confirmed species changes and observed new invaders after the almost eradication of several woody species following the drought periods of the 1970s and 1980s.

The Eastern Sahel shows significant species changes and a turnover into more drought tolerant species. First results of the vegetation survey in Darfur/Sudan document a dominance of Sahelian species. A total of 9,332 trees/shrubs had been measured in all five Darfur states. The top 15 dominant ligneous species show the following distribution: eight out of 15 belong to the pan-Sahelian domain, three are typical eastern Sahel species, three are representatives of the Saharo-Sahelian zone and one is a representative of the Sudano-Sahelian subzone and also a key species in fallow systems which can be found in the entire Sudano-Sahelian zone. Historical documents clearly describe a mixture of Sudanian, Guinean (South Darfur) and Sahelian species (e.g. Harrison and Jackson, 1958; Ibrahim, 1984) in Darfur before the 1970s drought. During a field survey in 2014 mesic trees had only been observed in depressions and around ponds. In West-Darfur approx. 10-12 years old fallow land did not show any regrowth of mesic species. The most dominant trees/shrubs in fallows were *Guiera senegalensis* and *Boscia senegalensis*, both pioneers and typical representatives of succession states. Figure 10 shows residuals of *Dalberghia melanoxylon* (Sudanian species) in the already highly fragmented savanna in West Darfur. Development of biological soil crusts is a typical phenomenon in desertification processes (Hahn and Kusserow, 1998).

**Figure 10.** Residuals of initially dense *Dalberghia melanoxylon* communities.

As one part of the still ongoing Darfur project additional information on natural resources conditions was gained through interviews. ~~Four communities in four states were targeted (urban, rural, nomads and the internally displaced persons [IDPs]). In total 102 communities and 2,547 households in Darfur had been interviewed during May to October 2014. Group discussion (12-20 people) was used as method for data collection. The interviewed persons were between 30 and 50 years old.~~ The findings confirmed an increase of more drought tolerant species at the expense of Sudanian species and an increase in bush fallow pioneers like *Guiera senegalensis*.

Contributions of other authors: When they compared post-drought with pre-drought conditions, several authors observed a species change since the beginning of the millennium. This is in line with findings from questionnaire-based surveys among elder people in the Sahel (e.g. Rasmussen et al., 2001; Gonzales et al., 2004, 2012; Ouedraogo et al., 2010; Brandt et al., 2014c; Sambou et al., 2016) documenting a clear increase of robust woody species at the expense of more mesic species. The survey period covered at least 20 years up to more than 40 years.

For Senegal, Gonzales (2001) analysed vegetation density and composition on the basis of historical literature and confirmed a shift to more aridity-indicating species (original Sudanian species were replaced by Sahelian species) as well as significant declines in vegetation density. For two sites in the Ferlo/Senegal, Vincke et al. (2010) observed a general shift towards more Sahelian and more sclerophyllous species of less socioeconomic importance and a general decrease of Sudanian species. Brandt et al. (2014c) reported the prevalence of few robust species (*Balanites aegyptiaca, Combretum glutinosum, Acacia raddiana*), which make up approx. 80% of all ligneous vegetation surveyed for a site in Senegal. Herrmann and Tappan (2013) found a reduction of woody species richness and a shift to more xeric species since the early 1980s, as well as an increasing dominance of shrubs in central Senegal. Other studies from Senegal and Mali confirmed these results (Maranz, 2009; Miehe et

al. 2010; Herrmann and Tappan, 2013; Brandt et al., 2014b, 2015, 2016a; Spiekermann et al., 2015). Kaptué et al. (2015) refers to a post drought tree population recovery and attested a decline in populations of economically and culturally important trees and shrubs despite the increase in woody cover. For Burkina Faso (Oursi-dune), Rasmussen (1999) noted that the species currently invading the live dunes - created in the 1970s - are not the

5 same as those dominating before. Analyses of neighbouring Gourma region in Mali confirmed these findings (Hiernaux et al., 2009a). Rasmussen et al. (2001) documented a significant increase of *Balanites aegyptiaca* on the basis of group interview with Peulh pastoralists. Wezel (2004) observed a decline of economically important trees, as well as a decrease in species which in drought periods are absolutely essential for survival like *Boscia senegalensis* for sites in Burkina Faso, Niger and Senegal.

Gonzales et al. (2004, 2012) investigated changes in forest species on the basis of interviews and field observations in 14 villages across the Sahel (5 states). The observation period included forty years (1960–2000). They found a significant decrease in forest species richness and discussed a shift of Sahel, Sudan and Guinea vegetation zones.

An increase in bush fallows with prevailing *Guiera senegalensis* and *Combretum glutinosum* had been

observed in Bandiagara (Brandt et al., 2014a) and in Ferlo, Senegal (Vincke et al., 2010; Brandt et al., 2014a, Brandt et al., 2017a).

The increase of ecological key indicator species (Miehe et al., 2010) like *Balanites aegyptiaca, Acacia raddiana* accounts for a fundamental change in the former Sudano-Sahelian ecozone. Trochain (1940) already discussed two forms of substitution of the original flora in Senegal:

(1) "Paratype of substitution" *Balanites aegyptiaca* is dominant

(2) "Savane – garrique anthropozoogène" *Combretum glutinosum* is dominant

An overview of plant species changes according to authors and region is given in Sect. 7 (Fig.18). The observed changes in species composition in the Sahel clearly indicate an impoverishment of an originally much more diverse flora. The parkland system which dominates large parts of the Sahelo-Sudanian zone is most likely

of human origin (Schulz and Pommel, 1992; Maranz, 2009; Schulz et al., 2009). Parklands are fundamental for livelihood in the Sahel-Sudan region. Low annual crop yields are often offset by fruit yields from trees maintained in parklands. As such they play an important role for local subsistence and enable the rural population to better overcome dry periods (Krings, 1991b; Maranz, 2009). The latter author found a remarkable decline in biodiversity including losses of species of Sudanian and Guinean provenience, associated with an

increase of Sahelian species. He discussed the widely observed senescence and disappearance of mesic species as a collapse of an anthropogenic system that is no longer adapted to increasing arid conditions due to ecologically critical rainfall shifts.

Ibrahim and Schulz (in press2017) investigated a sediment core from the Guidimouna-lake in SE Niger. Based on sedimentology, micromorphology and high-resoultion pollen diagrams, they were able to reconstruct the lake

history during the last 90 years. The author discussed the 1970s drought as the key trigger for plant species composition change favouring more drought tolerant species. They concluded, that the system could not recover completely yet, even regeneration of vegetation and soil is still present.

An additional indicator for the shifting ecosystem is wildlife. In Niger Giraffes, that were described to be present in the 1970s in the region of Tillabéri close to the border to Mali (Bernus and Hamidou, 1980), are now

found in the area of Dosso southeast of the capital Niamey (own observations), thus indicating a southward shift of approx. 150 km. Reichelt (pers. communication) reported the occurrence of giraffes and elephants even in the Gourma area in Mali at the end of the 1950s during his numerous geological field surveys (Reichelt, 1972).

It would be essential to continue field work based studies as mentioned by several authors (e.g. Miehe, 2010; Brandt et al, 2017a). However, the actual security situation in the Sahel especially in Mali, Burkina Faso and

45 Niger – regions like Gourma in Mali, Oudalan in Burkina Faso and almost the entire Niger are insecure (Weiss, 2016) – makes a reliable field survey planning impossible.

Positive developments are farmer-managed natural regeneration of selected trees on fields, reported for south-eastern Niger (Larwanou and Saadou, 2011; Sendzimir et al., 2011; Boubacar, 2016; Herrmann and Sop, 2016). In a recent study Brandt et al. (2017 a) observed an increase of *Pterocarpus lucens* (sudanian element) in those

50 parts of eastern Senegal that are characterised by less human pressure. The authors discussed low human impact as a key factor for regeneration.

**4.35 Re-greening Sahel based on NOAA-AVHRR, GIMMS3g, MODIS, SPOT VGT studies**

One key argument in recent scientific papers dealing with the question of Sahelian re-greening is based on NDVI analysis of coarse resolution satellite images. Given a concrete example, the author discusses limitations, when using post-drought satellite data.

Own contributions based on a multi-line approach in the Canal du Sahel area (southern Mali, close to the Mauritanian border) show a re-greening trend for the research site when comparing Landsat data, recorded in 1985 and 1991. However a comparison of data, received in 1976 with 1991 show the opposite trend –a significant decline (Figs. 11 and 12, Kusserow, 1994). The MSS false colour images (vegetation is red) of 1985 displays nearly no vegetation signal apart from the rice plantations of the Canal du Sahel recognisable in the

eastern part of the image (Fig. 11). However, the attached photograph of 1985 indicates dense woody vegetation cover. Due to the drought period (1982–1984) trees and shrubs were completely leafless. Many species in the area show different strategies to overcome the annually dry season without a lot of rain. The trees/shrubs either shed their leaves completely/partially (deciduous or semi-deciduous) or they are evergreen (Le Houérou, 1989).Typical aspects outside of drought periods are semi-deciduous coverage of ligneous vegetation, whereas

the foliage degree depends on tree/bush species and conditions of the previous rainy season. Is a drought period severe (as it was during the early1980s), woody vegetation shed their leaves completely. It appears that woody vegetation cover have been died off. The 1991 TM imagery depicts a recovery in woody vegetation cover. However ground survey indicated that some tree species, particularly Sudanian ecozone species as *Pterocarpus lucens* have not survived (Kusserow, 1994).

The NOAA-AVHRR sensors from that period may have also recorded dense woody vegetation cover without identifying them as woody vegetation because of its leafless state. Due to the strong precipitation deficit the herbaceous cover was hardly present. One could therefore argue that the observed re-greening since the early 1980s was predominantly based on the increase in agricultural crops and herbaceous cover.

**Figure 11.** Satellite imageries from the Canal du Sahel region in Mali show the situation during the drought period (February 1985, see also view from the ground, June 1985) and during the so-called re-greening (February 1991, size of subset is approx. 70 km x 50 km).

~~**Figure 12.** Same satellite subsets demonstrate a much denser savanna vegetation still in the 1970s, indicating a~~
~~post-drought browning.~~

In fig. 12 "re-greening" (period 1985-1991, top row) is compared with "browning" (period 1976-1991, bottom row). A legend provides information to identify individual land units. The MSS imagery recorded 1976 (bottom left) shows a dense and uniform woody vegetation cover (recognisable in the western part of the image by its brownish-reddish colour). The black patches are actual burnings. Between both dominating landscape elements

(uniform savanna and rice plantations) a small region with less or no vegetation can be discriminated. These areas are alluvial soils (light blue) and sandy areas/dunes (white-yellow, Kusserow 1994). Although the region had been experienced the drought period of the early 1970s, a uniform woody vegetation pattern still exists. The ecosystem resilience still seemed to be quite strong during the 1970s. The turning point showed up with the

renewed and more severe drought period from1982–1984. This decade can be classified as the starting point of a new ecosystem state.

Comparing the TM 1991 imagery with a MSS Landsat image from 1976 (bottom left ~~see Fig. 12~~) clearly indicates a significant decrease in woody vegetation cover which points to ~~is indicating~~ a browning trend.

**Figure 12.** "Re-greening" (period 1985-1991) and "Browning" (period 1976-1991) visible in satellite subsets of the Canal du Sahel region.

Contributions of other authors: to monitor vegetation trends in the Sahel, researchers used v~~V~~arious versions of ~~the~~ NOAA-AVHRR ~~(National Oceanic and Atmospheric Administration - Advanced Very High Resolution~~
~~Radiometer) data have been used to monitor vegetation trends in the Sahel~~ (e.g. Tucker et al. 1991, 1999; Anyamba and Tucker 2005; Herrmann et al., 2005; Dardel et al., 2014b; Knauer et al., 2014) and ~~.~~ Global Inventory Modeling and Mapping Studies (GIMMS; ~~)~~ ~~with a very coarse spatial resolution (5-8 km,~~ Mbow et al., 2015, Rasmussen et al., 2015)~~ are applied~~. On the basis of ~~Based on~~ these studies, scientists postulated a re-

greening trend in the Sahel since the early 1980s (e.g. Anyamba and Tucker, 2005; Herrmann and Hutchinson, 2005; Herrmann et al., 2005; Olsson et al., 2005; Helldén and Tottrup, 2008; Hiernaux et al., 2009b; Nicholson et al., 2012; Anyamba et al., 2014; Fensholt et al., 2013, 2015, Brandt et al., 2016a, b, 2017a).

~~New satellite derived imageries such as the new generation GIMMS-3g with a coarse resolution factor of 8 km; MODIS (Moderate Resolution Imaging Spectroradiometer) available since 2000 with a spectral resolution of 250 m and „SPOT Vegetation (VGT) with a resolution of 5 km and 1km (since 1999) are used more recently (Anyamba et al., 2014; Brandt et al., 2014a; Mbow et al., 2014; Rasmussen et al., 2014). Fensholt et al. (2004, 2015) and Brandt et al. (2014a, d) used datasets from MODIS, Geoland GEOV1 (5 km resolution) and GIMMS3g (8 km resolution) FAPAR (Fraction of absorbed photosynthetically active radiation) to assess local vegetation trends in Senegal and Mali. Horion et al. (2014) explored how dry season NDVImin can be used as proxy indicator for assessing changes in tree cover density.~~

Various authors mentioned limitations assessing vegetation changes (re-greening) when using coarse resolution NOAA-AVHRR and similar data sets, for instance:
- Data are only available for a relatively short period of time of approximately 30 years and only after the severe drought period of the 1980s (Gonzales et al., 2012; Mbow et al., 2015).
- Degradation is rarely visible with 8 km resolution from NOAA-AVHRR. With MODIS, SPOT VGT and other high resolution imageries areas with less vegetation cover, often hidden in GIMMS pixels, are clearly identifiable (Rasmussen et al., 2014; Mbow et al., 2015; Herrmann and Sop, 2016). This is confirmed by Brandt et al. (2014a) who identified degraded areas on the basis of coarse Geoland GEV1 FAPAR data, whereas GIMMS based pixel mixed degraded areas with agriculture. According to Brandt et al. (2014d) the choice of the datasets has significant impact on the result.
- Either the degradation processes are spatially very limited and thus the resolution of the existing sensors are not sufficient enough or the degradation is not strong enough to be discernible by remote sensing methods (Miehe et al., 2010; Dardel et al., 2014b).
- According to Brandt et al., (2014b) crusting surfaces are not detectable at a scale of 5 km and sometimes not even at 250 m (MODIS).
- Areas of farmer-managed natural regeneration in southern Niger, where field tree cover is said to have improve, do not stand out in satellite-derived greenness trends (Hermann and Sop, 2016).

In their review article on greening trends in the Sahel/Sudan, Knauer et al. (2014) noted that the observed trend is obviously due to various causes and can be interpreted as improvement but also as degradation. Herrmann and Sop (2016) concluded that long time series of NDVI has proven insufficient for detecting the woody fraction in semi-arid environments. Bachmann et al. (2015) emphasized the importance of a consistent pre-processing and harmonisation of the generated AVHRR time series.

Increase in greenness could also be caused by agricultural crops and grasses (Gonzales et al., 2012; Brandt et al., 2014a; Dardel et al., 2014a, Hiernaux et al., 2016). NDVI variability and trends are predominantly linked to herbaceous cover dynamics. Due to the influence of peak rainy season (August and September) the radiometric response of a woody plant cover is hardly distinguishable from an herbaceous cover (Dardel et al., 2014a). The authors concluded that an increase/decrease of woody vegetation cover could not be detected when using NDVI data from peak season at 1 km scale or larger. Even more important is that potential changes in the woody vegetation cover are not easily linked to the overall re-greening trend, because the re-greening is mainly linked with herbaceous and agricultural productivity (Dardel et al., 2014b). According to Bégué et al. (2011) who analysed NOAA-AVHRR for a period of 25 years, an increased cropping intensity is responsible for an increase in the annual NDVI for the Sahelian part of the Bani catchment area in Mali.

Many ~~Various~~ authors confirm large increases in agriculture (Ouedraogo et al., 2010; Knauer et al., 2014; Herrmann and Sop, 2016). During a 25-year period, Brink and Eva (2009) investigated the changes in forest, natural non-forest vegetation, agriculture and barren land in Sub-Sahara Africa (SSA) using high spatial resolution Earth observation satellites (Landsat). For the whole SSA the Sudanian eco-region shows the highest increase in agriculture (26%). The losses in non forest vegetation are also dominated by the Sudanian region (36%), followed by the Sahel region (29%). Compared to 1975, the Sudanian zone has also significant increases in barren lands (26%, Brink and Eva, 2009). Knauer et al. (2017) found that 91% of the agricultural area in Burkina Faso has expanded between 2001 and 2014. The expansion of agricultural land is also a result of the increasing population – Sahel countries have the worldwide highest population growth rates – and aggravate not only the strain on natural resources but also the already existing conflicts between agriculturalists and

pastoralists (Müller et al., 2011; Brücher et al., 2015). With the expected increase of the Sahelian population from about 13~~4~~2 million today to 197~~200~~ million in the year 2030 and 3~~26~~45 million in 2050 (DWD, 201~~7~~6), food demand, expansion of agricultural areas and demand of wood resources (fire wood) will increase dramatically.

~~Recent studies include biophysical variables like FAPAR and LAI (leaf area index), seasonal vegetation dynamics and land surface phenology (Ivits et al., 2013; Brandt et al., 2014a, d; Fensholt et al., 2015; Gessner et al., 2015; Diouf et al., 2015, 2016). Brandt et al. (2016a, b) apply a phenology driven model to estimate woody canopy cover in the Sahel at 1/0.5 km resolution scale on the basis of MODIS and SPOT-Vegetation FAPAR data. Based on several assumptions, their extrapolated woody cover map for the Sahel shows a site specific trend with areas documenting a positive development and areas with vegetation losses.~~ A phenology-driven model for estimating woody canopy cover in the Sahel at 1/0.5 km resolution scale on the basis of MODIS and SPOT-Vegetation FAPAR data were applied by Brandt et al. (2016a, b). Based on several assumptions, their extrapolated woody cover map for the Sahel shows a site specific trend with areas documenting a positive development and areas with vegetation losses. The authors concluded an overall positive trend in woody cover, emphasising the resilience of the ecosystem. In a further study carried out by Brandt et al. (2017a) the authors found a significant spread of mostly *Combretaceae* and *B. aegyptiaca* for pastoral areas of central and eastern Senegal. For Africa's drylands Brandt et al. (2017b) discussed positive changes in woody vegetation cover with climate and $Co_2$ as the main drivers. This woody cover increases, however, were found to be offset by human impact (logging and agricultural expansion).

New approaches to meet the challenge of NDVI-based separation of fields and fallow is given by Tong et al., 2017, who investigated the coupling between NDVI trends and cropland changes for a test site in western Niger (Fakara). The authors found positive NDVI trends based on more frequent fallow years and a negative NDVI trend associated with an increase in cropped fields. Mbow et al. (2013) stressed that major changes in plant species dominance should be taken into account when analysing NDVI time series.

## 4.4~~6~~  Changes in vegetation pattern, self-organised patchiness

Own contributions: Significant changes in the spatial distribution of woody vegetation were first observed when using multitemporal remote sensing data from the early 1950s to 1992 (aerial photographs, Landsat, SPOT) for vegetation monitoring in Mali (Kusserow, 1990, 1994). Within 40 years the originally homogeneous vegetation pattern, still recognisable in the 1950s (example Mali, Canal du Sahel region) has turned into a highly fragmented pattern with spots and isolated bands. The main trigger for this development was found to be human impact (clearing and wood cutting). Derived from these remote sensing based observations, a principal model (Fig.13) for the development of woody vegetation patterns towards desertification was developed (Kusserow, 1994).

**Figure 13.** Left: Changes in the spatial vegetation distribution from North to South: Sahara – South Sahara/North-Sahelian transition zone – Sahel (modified according to Kusserow 1994). Right: Chronology of changes in spatial vegetation distribution in the south, i.e. Sahel (modified according to Kusserow 1994).

The changes in the spatial woody vegetation distribution from North to South (Fig. 13, left) correspond with the rainfall gradient from the arid Sahara to the more humid areas in the south, i.e. from the desert ecosystem to the savanna ecosystem. The same pattern changes but in reverse order were observed in savanna areas (Sahel) when analysing aerial photographs and satellite data from the 1950s/1970s until today (Fig.13, right). Referring to Monod (1954) who defined the approximate borderline between Sahara and Sahel as one between contracted (Fig.13 upper left picture) and scattered-diffuse vegetation types (Fig. 13 bottom left), the author discussed these observed changes as a principal indicator for desertification.

Gaps, stripes and spots as detectable in the 1975 aerial photograph had been later postulated by several model studies (e.g. Lejeune et al., 2002; Dekker et al., 2007; Gilad et al., 2007; Meron, 2012). On the basis of freely accessible Landsat series of the Canal du Sahel region, the development from 1976 until 2010 could be continued. The comparison of two Landsat imageries (size of subset is approx. 80 km x 55 km) recorded on 26 February and in different years (1976 and 2010), demonstrates a transition from an originally uniform woody

vegetation pattern into a banded and spotted distribution within a time span of 34 years (Fig.14). A legend helps to identify individual land units and principal land cover changes. Main emphasis is laid on the process of vegetation pattern development. The false colour images of 1976 still shows a dense and uniform savanna vegetation in the western part of the imagery. The corresponding image of 2010 documents significant changes: the originally dense and uniform savanna vegetation has disappeared and a fragmented, discontinuous woody vegetation distribution has emerged (banded and spotted pattern in the western part of the image). The topographic map from IGN (Institute Géographique National Paris, République du Mali_Feuille ND-29-XVIII, IGN Paris, February 1961) still documented a tiger bush in the 1950s in this area.

**Figure 14.** Development of vegetation pattern in the Canal du Sahel region in Mali over a period of 36 years (1976–2010, size of subset is approx. 80 km x 55 km).

The actual burnings are restricted to the dune systems in 2010 (visible in the lower half and the uppermost part) whereas the upper half indicate no burning but significant vegetation pattern formation (western part of the imagery). As discussed further above, regeneration can be documented on sandy soils. The region between both dune systems is characterised by shallow soil cover (Ferric luvisols, Di Bernardo et al., 1986) over Precambrian basement. These areas show none or only little agricultural activity. The shallow soil layers are extremely vulnerable to degradation and desertification processes (see Sect. 4.13). The development of woody vegetation pattern (contraction), clearly visible in the 2010 imagery, started exactly from these areas. Analyses of additional Landsat time series for the test site in Mali (not presented here) document the starting of fragmentation processes (pattern formation) after the 1980s drought. The imagery recorded in 1991 does not yet show any significant structures, whereas at the image dated 1999 pattern formation has already started to form. This pattern is a lot more clearly identifiable from 2001 on and much more significant from 2010 on. Based on interpretations of images from the Landsat satellite image archive, the process of pattern development was estimated to have been completed within 10 to 15 years (example Mali).

The second example is located in western Niger, north of Ouallam and close to the border of Mali (Fig. 15). The size of the area is around 60 km x 45 km. Two main landscapes are recognisable: lateritic plateaus (dark green) and dune systems (yellow-white) partly covered by vegetation. The focus is on the process of woody vegetation pattern development (fragmentation/contraction). Identification of individual land units is supported by a legend. The image recorded on 30 September 1973 shows the impact of the 1970s drought period, agricultural activities are hardly detectable. The annual precipitation is low (approximately 290 mm). The average rainfall amount for the period 1972–2001 is 366 mm, indicating a typical Sahelian environment. Rainfall data originated from the weather station Ouallam (Meteorological service in Niger, 1973–20029; from 200310 on complemented by GPCC V5-data). The data were processed by Andrea Oestreich, Meteorological Institute, Freie Universitaet Berlin.

Uniform woody vegetation (violet-purple) is still dominant in 1973 (30 September). For the early 1960s the occurrence of bush and tree savannas was confirmed by IGN topographic maps (sheet Ouallam, République du Niger Feuille ND-31-XV, IGN Paris, March 1961). The second image was recorded 40 years later on 27 September 2013 and presents a higher rainfall situation indicated by a dense herbaceous vegetation cover and agricultural activity on the dunes. Although the precipitation dept is higher (4117 mm) compared to 1973, the vegetation cover appears in a reversed order, i.e. vegetated dunes are recognisable but bare areas and fragmented woody vegetation patterns had been developed on the argillaceous sandstones of the Continental Terminal (Greigert and Pougnet, 1965). Parts with bluish colours (former vegetated areas) are now representing degraded, mainly crusted soils.

**Figure 15.** Comparison of two Landsat imageries of 1973 and 2013 showing how ligneous vegetation pattern formation has formed out of an originally uniform vegetation cover within 40 years (size of subset is approx. 60 km x 45 km).

The woody vegetation patches observable in the satellite image series (Fig.15) emerged in the valley bottoms (sheet Ouallam, IGN, 1961). Due to increasing run off towards the valleys (Mainguet, 1991; Kusserow, 1994) vegetation re-growth has improved. At the same time, accessibility via roads has worsened, so that roads relocated from the valley bottom (still shown in the topographic map) to the lateritic plateau (in situ

observations). The following Landsat satellite based time series depict ~~show~~ the forming of woody vegetation pattern (Fig. 16). Subsets of six satellite images show principal changes in vegetation distribution (please note that Landsat imageries, recorded 30 Sep 1973 and 19 Sep 2016, are only shown as subset).

The imagery recorded 13 October 1984 already presents a significant pattern of stripes and spotted areas with some residuals ~~relies~~ of a more uniform distribution (rainfall amount in 1984: 160~~75~~ mm). Due to the drought period of the early 1980s the dunes show less vegetation cover and hardly agricultural activity. The image recorded on 3 October 2002 (high rainfall with 48~~9~~0 mm) shows that dunes are re-vegetated and agricultural activity (fields) is clearly recognisable. In the upper northern part no regeneration is detectable but pattern formation (stripes and spots) of woody vegetation is well visible. During a field trip in April 2001, farmers in the small village Tuizégourou complained about harsh environmental conditions and low agricultural productivity with increasing risk of crop failure. According to local people these areas were starting points for emigration. The region has a longstanding settlement history which dates back until the late Neolithic period. Devisse and Vernet (1993) proved settlements between 2000 and 500 B.C. by means of C14 dating method. The 2009 (18 October) imagery depicts the area after a rainfall shortage (29~~8~~0 mm) comparable to the drought in 1973. Pattern formation is clearly recognisable. The 2014 image presents the effects of higher rainfall (413 mm) with vegetated dunes and fields. Significant woody vegetation pattern is clearly visible. The presented subsets document the principal process of pattern development which can be used as tracing marker for ecosystem changes. The image sequence clearly indicates that vegetation patches are stable once they are established, despite an increase in annual rainfall. On the basis of mathematical models of vegetation growth, Rietkerk et al. (2011) reported that increased rainfall may not recover the spotted state because the resource concentration mechanism (concentration of soil water under vegetation patches) fails.

**Figure 16.** Pattern sequence of a research site in western Niger (North of Ouallam).

For the research side in West-Niger the pattern formation process was observed to have been occurred between 1973 and 1984 (two drought periods) which would be a very short formation period of 11 years. Pattern changes are triggered by increasing aridity and exacerbated by human impact. This development is mainly observed on shallow soils (see Fig. 14-16).

Based on time series of Landsat MSS/TM and SPOT satellite images, aerial photographs and kite photographs, Kusserow and Haenisch (1999) analysed the dynamics of a tiger bush site southeast of Niamey, Niger. They found that the banded patterns were formed out of an originally uniform state in the 1950s and interpreted vegetation stripes as a relic habitat or kind of biodiversity pool. Vegetation bands, soil sealing and crusting between bands form a surface layer protecting the seed bank (Hahn and Kusserow, 1998), thus constituting a crucial part of a natural in situ conservation strategy (Kusserow and Haenisch, 1999).

First results of an ongoing project in Darfur/Sudan show distinct vegetation pattern changes when comparing Landsat MSS data (spatial resolution 80 m) from the early 1970s and data from an Indian Microsatellite system (spatial resolution 37 m) recorded in 2010. Unlike the situation in the western Sahel, tiger bush areas could not be found, which is mainly due to a different geomorphology. Vegetation distribution changes can also be identified. The origin is similar: woody vegetation pattern formation has formed out of an originally uniform vegetation cover still recognisable in the 1970s satellite imageries. For all own research areas presented above, a strong human impact had been identified as main driver of the pattern development. As already mentioned, the author discusses vegetation pattern formation as observed in satellite time series as a key indicator for desertification processes. The driving force is a feedback between drought and increasing human intervention, i.e. wood cutting and clearing for cropping.

Contributions of other authors: Special vegetation mosaics known as "tiger bush" (brousse tigrée) are common vegetation pattern in dry regions. These patterns consist of bushy stripes and arcs alternating with open, non vegetated areas that are often crusted, and situated on very gentle and uniform slopes. The type of pattern can vary between "spotted", "broadly" and "horizontally banded" (following the contours) (White, 1970; d'Herbès et al., 1997; Hiernaux and Gérard, 1999; Valentin et al., 1999). Due to the striped appearance on aerial photographs this phenomenon was called "brousse tigrée" by Clos-Arceduc (1956). Its occurrence is reported for the Sahel (White, 1970; Janke, 1976; Cornet et al., 1992; Thiéry et al., 1995; d'Herbès and Valentin, 1997; Hiernaux and Gérard, 1999) and other semi-arid regions (Valentin et al., 1999).

Several authors described the main formation mechanism: due to a better water balance in the upper soil (generated by sheet run-off on the bare inter-bands), the self-modifying system of vegetation stripes offer more demanding species the possibility to survive in habitats with less rainfall (White, 1970; Cornet et al., 1992; d'Herbès et al., 1997). D'Herbès and Valentin (1997) and Valentin et al. (1999) discussed the Niger tiger bush as a natural water harvesting system. According to their findings, the mean annual water infiltration into the thicket cores of vegetation bands enables wood production similar to that of woodland and forest in the wet savanna zones and even exceeded forestry industrial plantations.

~~Kusserow and Haenisch (1999) analysed the dynamics of a tiger bush site southeast of Niamey, Niger, on the basis of time series of Landsat MSS and TM and SPOT satellite images, aerial photographs and kite photographs. They found that the banded patterns were formed out of an originally uniform state in the 1950s and interpreted vegetation stripes as a relic habitat or kind of biodiversity pool. Vegetation bands, soil sealing and crusting between bands form a surface layer protecting the seed bank (Hahn and Kusserow, 1998), thus constituting a crucial part of a natural in situ conservation strategy (Kusserow and Haenisch, 1999).~~

Thiéry et al. (1995) discussed that the two common hypotheses – degradation of an initially uniform pattern or colonisation of previously bare zones – are two aspects of the same phenomenon.

These field and satellite based results were later confirmed by mathematical models of vegetation growth (von Hardenberg et al., 2001; Rietkerk et al., 2004). Recent studies using more advanced modelling techniques discuss this phenomenon as characteristics of landscapes with water-limited systems. Mosaics of patches differ in resource concentration, biomass production and species richness (Gilad et al., 2007). They are a key factor in driving ecological processes at different spatial-temporal scales and modifying vegetation distribution and species diversity, and may contain information on desertification processes (Von Hardenberg, 2010). Two types of vegetation patchiness in water-limited systems are discussed: a periodic pattern and an irregular scale free pattern; the latter one is more common in nature (Von Hardenberg et al. 2010; Kletter et al., 2012).

Rietkerk et al. (2004) highlighted the importance of two processes that attracted considerable attention in scientific community during the past decade: "ecosystem engineering" and "self-organized patchiness". Ecosystem engineers (Jones et al., 1994, 1997; Gilad et al., 2007; Meron, 2012) are organisms that modify, maintain and create habitats by causing physical state changes in biotic or abiotic materials and as such provide habitats for other species. Self-organised patchiness means a mechanism of positive feedback between plant growth and availability of water (Valentin et al., 1999; Rietkerk et al., 2004). As indicated by mathematical models, vegetation patterns~~s~~ are clearly related to an instability of spatially uniform vegetation (Thiéry et al., 1995; von Hardenberg et al., 2001; Rietkerk et al., 2004). The models predicted three pattern states of vegetation according to the rainfall amount: (1) an uniform vegetated state (dry-subhumid), (2) an arid and semi-arid state and (3) an uniform bare (hyperarid) state. The models also predicted a possible coexistence of different stable states under the same rainfall conditions. The range of coexisting patterns and bare states determines the extent of irreversibility of associated desertification process (von Hardenberg et al., 2001). Gilad et al. (2007) summarised five basic vegetation stages along the rainfall gradient (Fig. 17):

- Uniform stages at high rainfall
- Periodic gap, stripe and spot patterns at decreasing rainfall and
- Bare soil at low rainfall

**Figure 17.** Model results show vegetation pattern development from an originally bare state into an uniformly distributed vegetation, reflecting system's ability of optimal self-organisation with respect to water resources (Gilad et al., 2007).

Rietkerk et al. (2004) reviewed studies that linked self-organised patchiness to catastrophic shifts in ecosystems. Such catastrophes are commonly attributed to the existence of two alternative stable states in ecosystems. The authors defined this as bistability (see also Thiéry et al., 1995). Increased resource scarcity leads to a spatial reorganisation (see also Cornet et al., 1992 and Valentin et al., 1999). According to the model, certain spatial structures may develop in real ecosystems that only arise when resource availability has decreased. Simulations showed that under growing aridity conditions bare spots merge into 'labyrinthine' stripes which subsequently become a bare matrix interspersed with vegetation spots. If these vegetated spots disappear a complete desert may occur (Barbier et al., 2006). Rietkerk et al., (2011) proposed the hypothesis that imminent catastrophic shifts in ecosystems can be predicted by self-organized patchiness (ecosystem engineering). In these

models the vegetation shifts catastrophically from a spotted state to a bare homogeneous state if rainfall is decreased beyond a threshold. It is worth mentioning that according to Rietkerk et al. (2011), increased rainfall may not recover the spotted state because the resource concentration mechanism (concentration of soil water under vegetation patches) fails. If this development is associated with a massive loss of ecological and economic resources, it wills effects human societies dramatically. The previously mentioned numerical modelling studies mainly discuss pattern formation triggered by natural phenomenon (rainfall, geomorphology). Human impact is mentioned but not investigated in detail (Gilad et al., 2007; von Hardenberg et al., 2010).

   Vincke et al. (2010) also observed a contraction phenomenon in the Ferlo region in Senegal where they documented an increasing shift of two robust species (*Boscia senegalensis* and *Guiera senegalensis*) from tops to depressions. The authors suggested that these changes may have contributed to the shift from homogeneous vegetation pattern into a patchy distribution of vegetation. Barbier et al. (2006) applied Fourier analysis to high-resolution remote sensing data in south-west Niger. The analysed aerial photographs covered a period of 40 years (1996 to 1956). According to their results the formerly homogeneous savanna has been dramatically changed into a spotted pattern. Protected areas showed less-spotted pattern than areas characterised by strong human impact. The authors discussed the observed spatial vegetation changes as potential indicators for climatic and anthropogenic constraints and underlined that the intensity of the patterning process during the observation period of forty years was exacerbated by human activities. This is in line with investigations of Barbier et al. (2006) in south-west Niger, who had analysed aerial photographs for the period 1956 to 2006. The forming of contracted vegetation was confirmed by Couteron et al. (1997) on the basis of aerial photograph data from 1955 and 1984 for a site in Burkina Faso.

## 57   Key messages and new aspects

The following key messages and new aspects can be summarised from the above discussions:

- Recovery of vegetation predominantly depends on soil types, as shown in Sect. 4.13. Own field- and EO based studies for Mali, Burkina Faso and Niger since the early 1990s prove a favourable situation on Quaternary dune systems whereas poorly developed soils (basement, gravel plains, pediments) and crusted soils show less or no regeneration (Sect. 4.13, 4.2 and 4.66). This is in line with recent studies of other authors. In addition and as a new aspect, the non-recovery processes on certain soil types could be assigned to geological structures, i.e. Precambrian basement and Mesozoic/Cenozoic sediments for the northern part of Burkina Faso, western and central Niger and parts of southern Mali (Canal du Sahel region). Parts of eastern Chad and Darfur allow for similar assignments. This implies that all soil types are more or less vulnerable except sandy soils of Quaternary dune systems. It can therefore be reasonable for further EO-based studies to include geological-geomorphological maps to better understand resilience, re-greening- and browning trends in the Sahel. In particular reforestation campaigns had a greater chance of success, if knowledge about the extension of Quaternary dune systems were utilised. Areas showing poorly developed or crusted soils would need special treatment and management (Kusserow, 2010).
- Plant species changes are of great importance for debating Sahel greening vs. Sahel browning as discussed in Sect. 4.2. Own research results from 1994, 1995, 1998 and 2014 prove a significant decline in woody species biodiversity and a dramatic increase of more aridity indicating species at the expense of more mesic ones for sites in Mali, Niger and Darfur/Sudan. This was confirmed by later studies (e.g. Gonzales 2001). A compilation showing vegetation trends in the Sahel according to author and region is given in Fig. 18. The spread of two to three species (*C. glutinosum, Guiera senegalensis and B.aegyptiaca*) as stated by all authors, is a clear indicator of a succession state and points towards a further decrease in already very limited ecosystem services. More than 90% of the entire Sahelian population (approx. 12017 million people, DSW, 20176) depend on wood and charcoal (IEA, 2014; Mortimore, 2016). The pioneer species *Combretum glutinosum* and in particular *G. senegalensis* with very slim branches definitely do not belong to the preferred wood supply species.

It can be concluded from the comparison of post-drought and pre-drought conditions that a significant species change into fewer and more drought tolerant species has occurred. This demonstrates how important the selected observation period is for the desertification debate. the selected observation period is.

- Analyses solely based on NDVI data argue for a Sahel re-greening while a broader approach with Landsat-based analyses does not (see Scet. 4.3~~5~~). Investigation in the Canal du Sahel area shows a re-greening trend for the research site if Landsat data recorded in 1985 and 1991 are used. If earlier Landsat data from 1976 are compared with those from 1991, however, the opposite trend is visible – a significant decline of ligneous vegetation (Kusserow, 1994, 1995). This example also points towards the key question: which observation period is best used for trend analyses? Due to the severe drought period in the early 1980s, trees and shrubs may have shed their leaves completely. Thus, the NOAA-AVHRR sensors from that period may have also recorded dense woody vegetation cover without identifying them as woody vegetation because of its leafless state. The NDVI based Sahel greening is therefore to be questioned with three main arguments: (1) The observed re-greening since the early 1980s seems to be predominantly based on an increase in agricultural crops and herbaceous cover (currently under review, see Sect.4.3~~5~~); (2) statements on the development of post-drought ligneous cover bear little significance due to the temporarily leafless trees and shrubs and (3) a comparison with satellite imageries recorded in the 1970s indicate much more dense woody vegetation cover.

- Changes in woody vegetation distribution in relation to self-organised patchiness can be used as key indicator for desertification processes as discussed in Sect 4.4~~6~~. Own investigations conducted in Mali in the early 1990s indicated that the spatial cover of woody vegetation has changed within a time span of 40 years. The woody cover, originally characterised by an uniform (scattered-diffuse) distribution pattern, has turned into a highly fragmented pattern with spots and isolated bands. This specific pattern formation was considered as a principal indicator for desertification (Kusserow, 1994). This first scheme could be confirmed by analysing sets of Landsat data for test areas in Mali, Burkina Faso and Niger (see Sect. 4.1~~3~~, 4.2, 4.4~~6~~). This is a new aspect, brought into the debate of Sahel greening/browning. It could be also shown, that the postulated (based on numerical modelling) pattern formation time span of 37 years (Gilad et al., 2007) can be substantially shorter. Own satellite based analyses (Sect. 4.1~~3~~ and 4.4~~6~~) show formation time spans of only 10 to 15 years (Mali site) and 11 years (Niger site). Large areas stretching from Oudalan in Burkina Faso until Liptako in western Niger and beyond didn't show any vegetation recovery, despite an increase in rainfall since the 1980s. The subsurface of these areas consists of Precambrian basement and Tertiary sediments and is characterised by the emergence of a pattern of bare spots having developed from an initial diffuse or homogeneous vegetation distribution. Poorly developed soils on Precambrian basement/Mesozoic/Cenozoic sediments (see Sect. 4.1~~3~~) are especially prone to vegetation pattern formation. Thus, also for this indicator geological maps could support identifying vulnerable areas. Within three decades (post-drought) the bare spots could grow and form large vegetation-free areas as shown in Sect. 4.4~~6~~. In contrast to this development, the intersecting Quaternary dune systems show vegetated and agricultural areas depending on the variance of the yearly rainfall amount. On the basis of selected Landsat data from 1973 to 2016, a key indicator for ecosystem changes could be identified. The image sequence (Fig.16) clearly indicates that vegetation patches are stable once they are established, despite an increase in annual rainfall. According to Rietkerk et al. (2011) increased rainfall may not recover the spotted state because the resource concentration mechanism (concentration of soil water under vegetation patches) fails. This assumption can be confirmed on the basis of remote sensing data for regions in Mali, Burkina Faso, Niger, and Darfur/Sudan.

**Fig. 18:** Shift from pre-drought mesic to post-drought xeric plant species according to authors and region.

The discussion above shows the importance of the selected observation period when debating Sahel greening vs. Sahel browning. In addition, the findings presented in this article argue for a new understanding of the process of desertification in the Sahel region rather than further accentuating the two contrary positions of a greening Sahel vs. a browning Sahel. The author suggests considering the Sahel as an ecosystem that changed from an originally "greener" state into a new more desert-like system. Main indicators are species turn over and vegetation pattern formation. The tipping point was the renewed drought period in the early 1980s.

Finally, two key questions shall be raised for further research and debate:
- Which observation period should be taken as default period to assess ecosystem changes? Do we want to be dependent on temporal limitations of methods?
- How should the stability of an ecosystem be evaluated? Should we discuss any plant spreads (even it is an indicator for degradation and losses in biodiversity) as a positive sign? What does that mean with regard to the worldwide highest population growth rates in the Sahel?

**Acknowledgements.** I am grateful to Brigitte John, University of Bayreuth, for supporting me with the acquisition of satellite images, and Andrea Oestreich, Freie Universitaet of Berlin, for providing me with meteorological data and rainfall analyses for almost 30 years. I would also like to thank my husband Christian for his fruitful comments and critical remarks. I am especially grateful to all both reviewers who helped condensing and streamlining the article with their critical comments.

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

45

## Figures

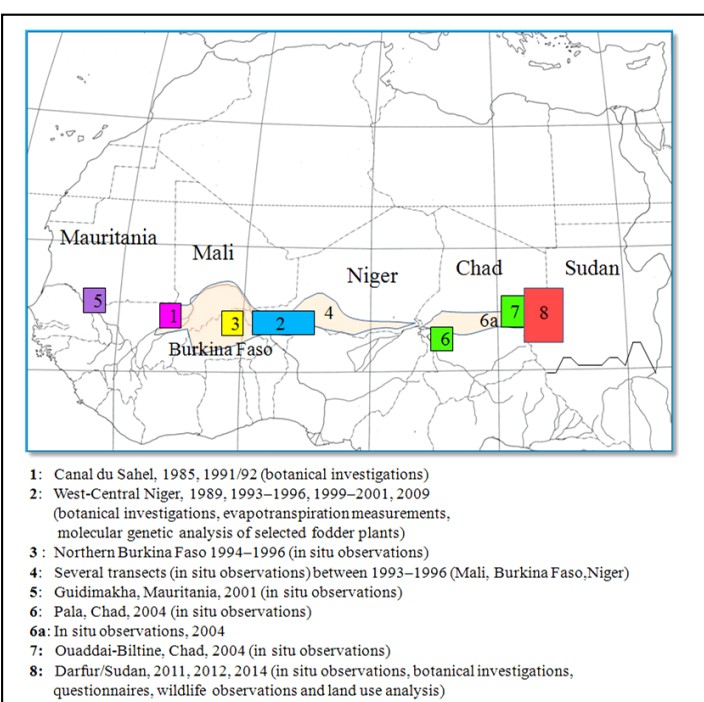

**1:** Canal du Sahel, 1985, 1991/92 (botanical investigations)
**2:** West-Central Niger, 1989, 1993–1996, 1999–2001, 2009
(botanical investigations, evapotranspiration measurements,
molecular genetic analysis of selected fodder plants)
**3 :** Northern Burkina Faso 1994–1996 (in situ observations)
**4:** Several transects (in situ observations) between 1993–1996 (Mali, Burkina Faso,Niger)
**5:** Guidimakha, Mauritania, 2001 (in situ observations)
**6:** Pala, Chad, 2004 (in situ observations)
**6a:** In situ observations, 2004
**7:** Ouaddai-Biltine, Chad, 2004 (in situ observations)
**8:** Darfur/Sudan, 2011, 2012, 2014 (in situ observations, botanical investigations,
questionnaires, wildlife observations and land use analysis)

**Figure 1.** Position of research projects and transects of observations.

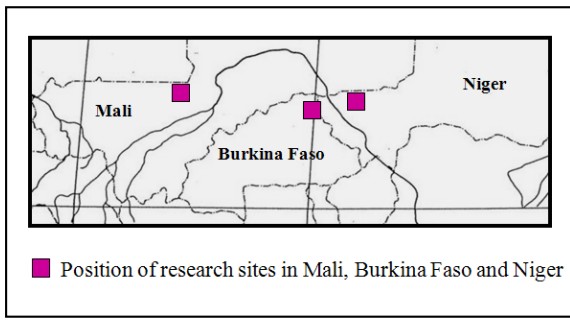

**Figure 2.** Position of research sites in Mali, Burkina Faso and Niger.

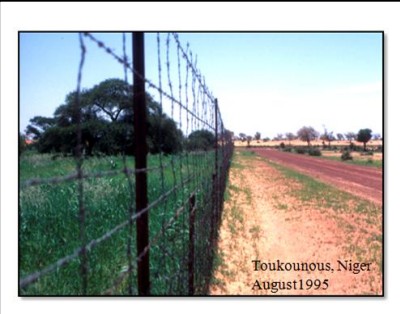
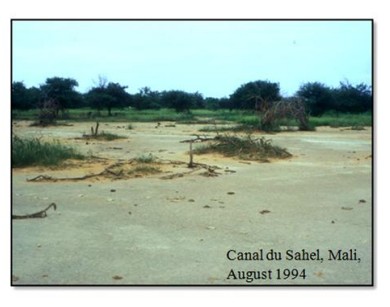

**Figure 3a.** Toukounous, Niger (August 1995).   **Figure 3b.** Canal du Sahel, Mali (August 1994).

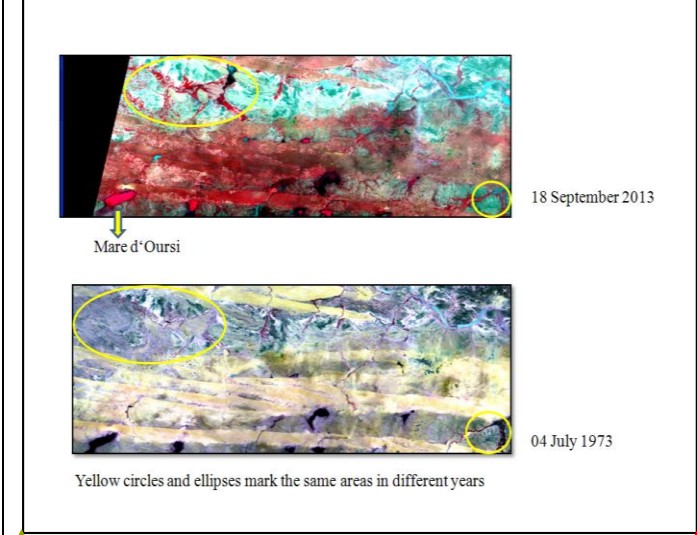

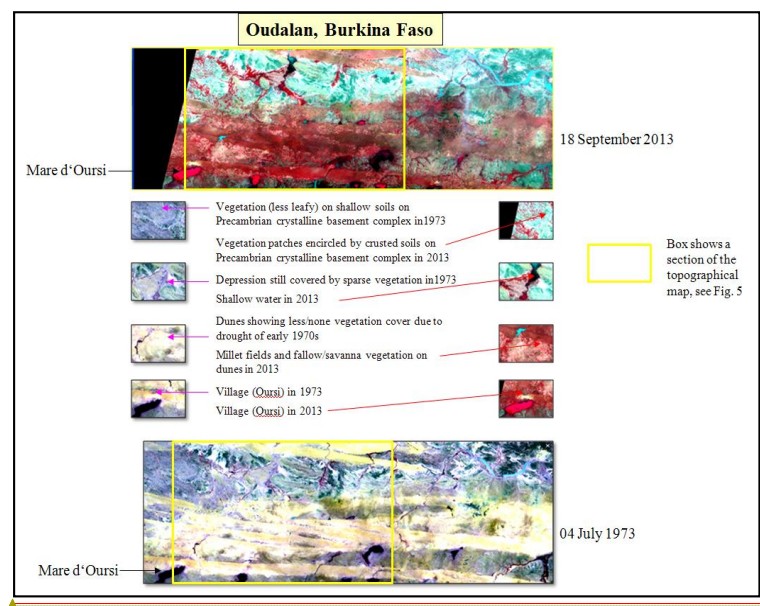

**Figure 4.** The Landsat subsets from eastern Burkina Faso cover an area of approx. 70 km x 30 km.

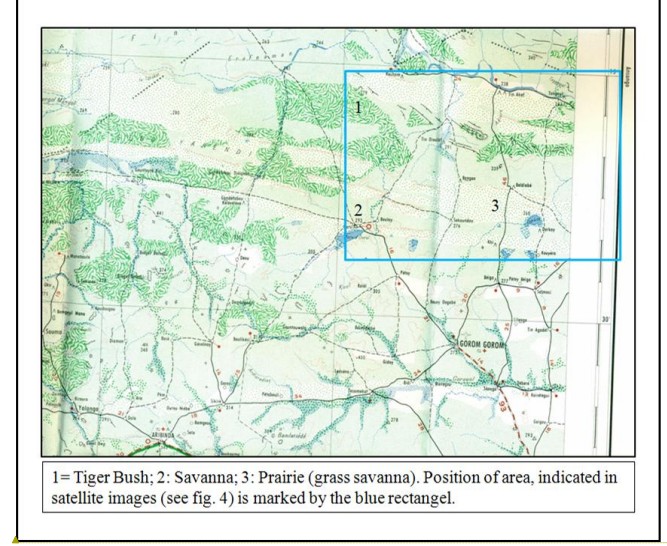

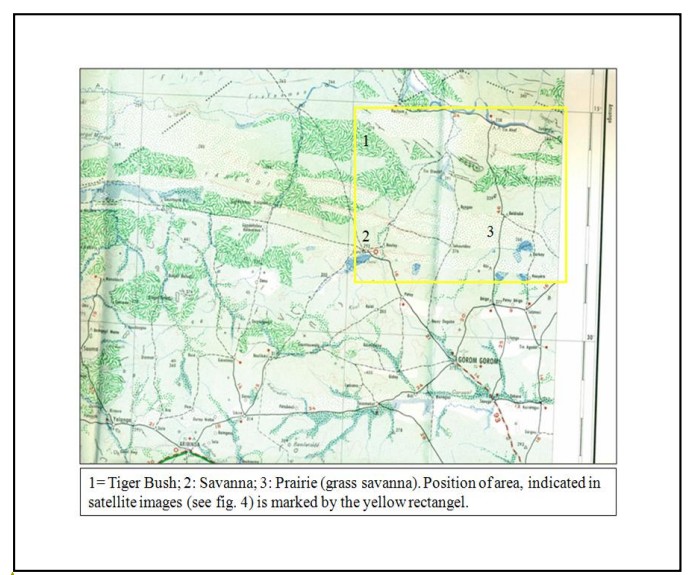

1= Tiger Bush; 2: Savanna; 3: Prairie (grass savanna). Position of area, indicated in satellite images (see fig. 4) is marked by the yellow rectangel.

**Figure 5.** Section of the topographic map "Hombori" (IGN 1961), indicating vegetation types of the early 1950s. 1: Tiger Bush; 2: Savanna; 3: Prairie (grass savanna). The box shows the position of the satellite image section (see Fig. 4).

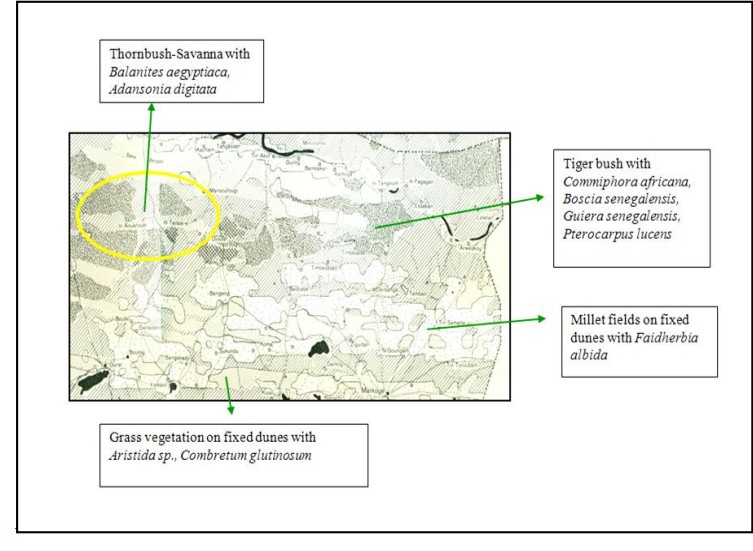

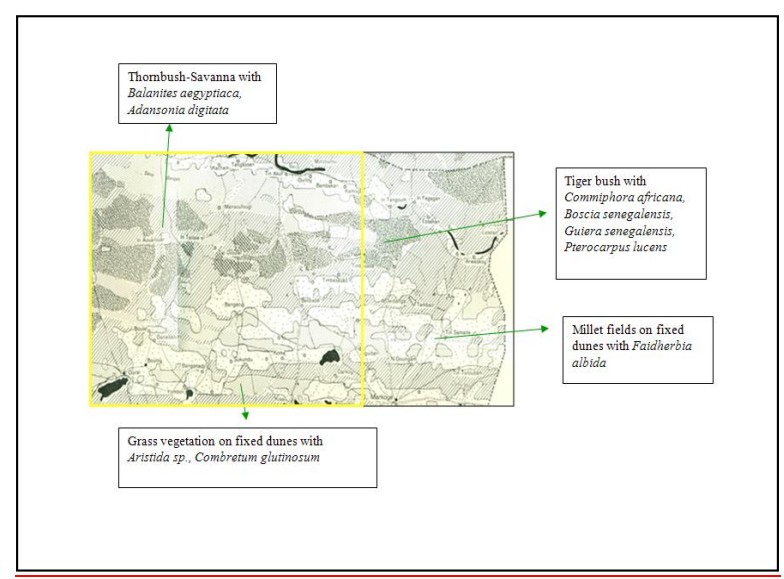

**Figure 6.** Vegetation and Landuse in the north-east of Upper Volta and lower Gourma, Mali, status 1976–1977, modified after Krings (1980). The box shows the position of the satellite image section (see Fig. 4).

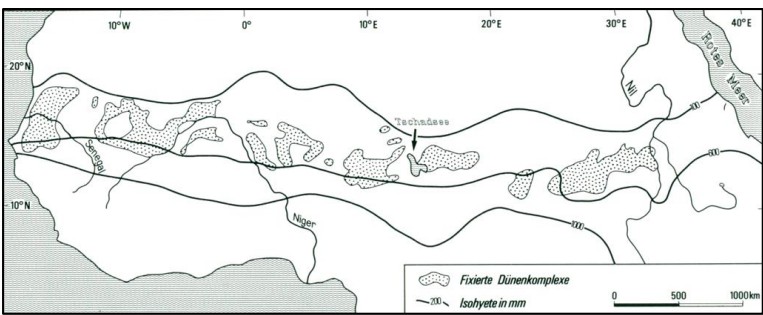

**Figure 7.** Position of late Quaternary dune systems in the Sahel according to Mensching (1990).

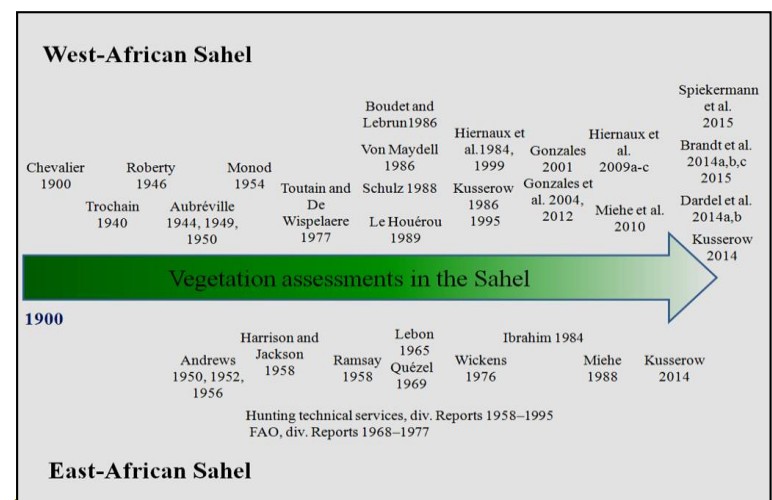

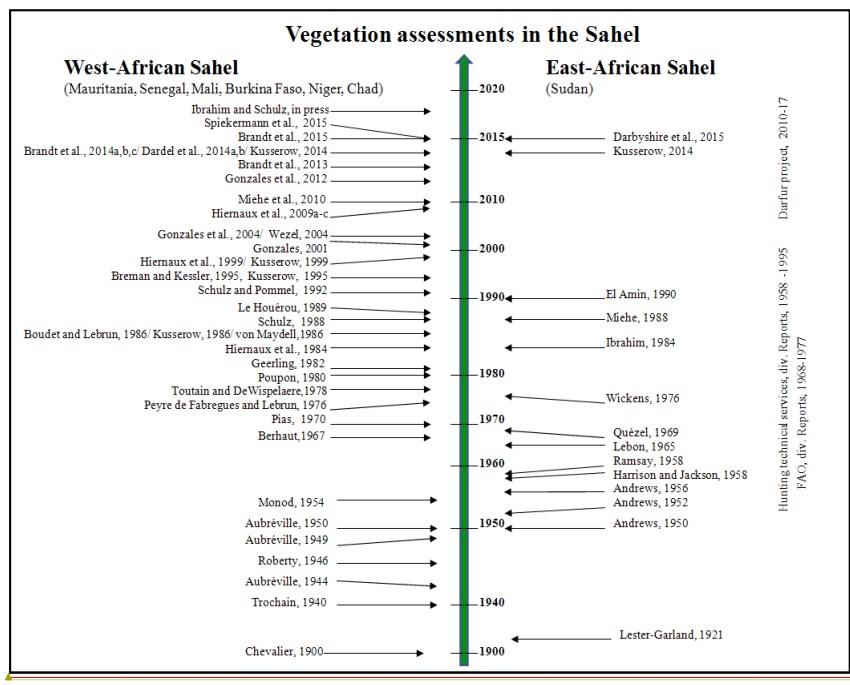

**Figure 8.** Compilation of selected botanical assessments in the West- and the East-African Sahel since 1900.

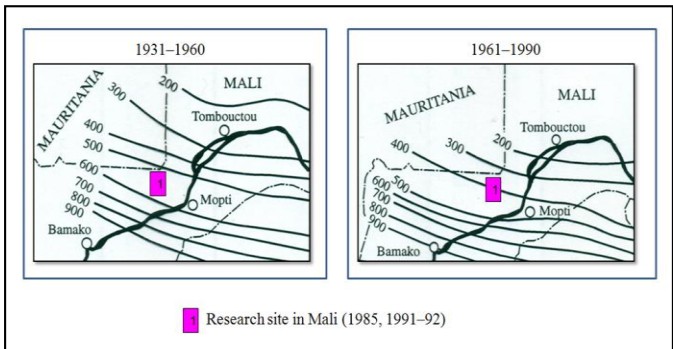

**Figure 9.** Location of average isohyets at CLINO 1 (1931–1960) and CLINO 2 (1961–1990), position of research areas is indicated by the red box (modified after Kusserow and Oestreich, 1998).

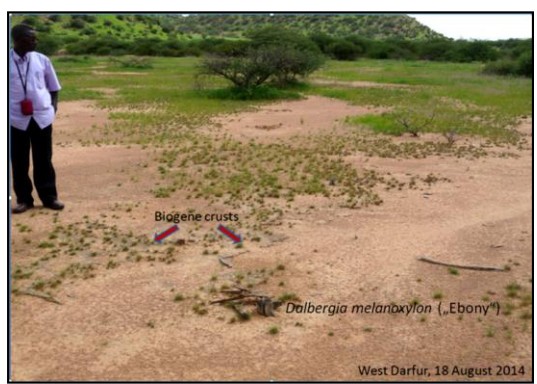

**Figure 10.** Residuals of initially dense *Dalberghia melanoxylon* communities.

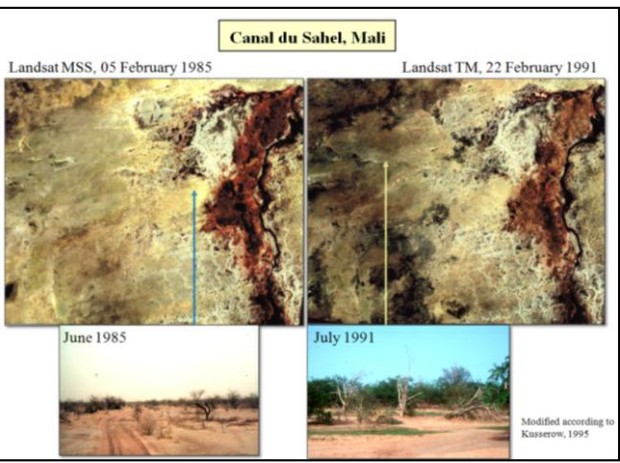

**Figure 11.** Satellite imageries from the Canal du Sahel region in Mali show the situation during the drought period (February 1985, see also view from the ground, June 1985) and during the so-called re-greening (February 1991, size of subset is approx. 70 km x 50 km).

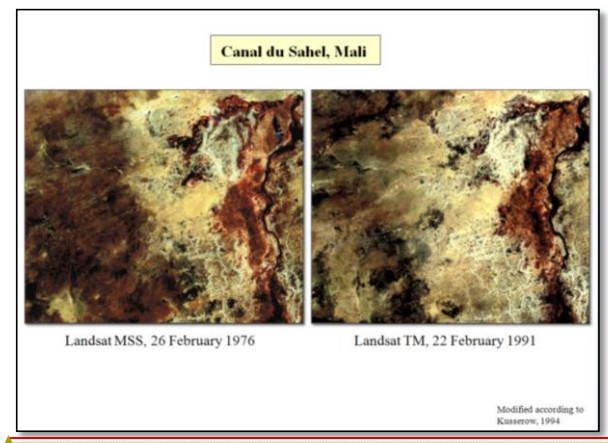

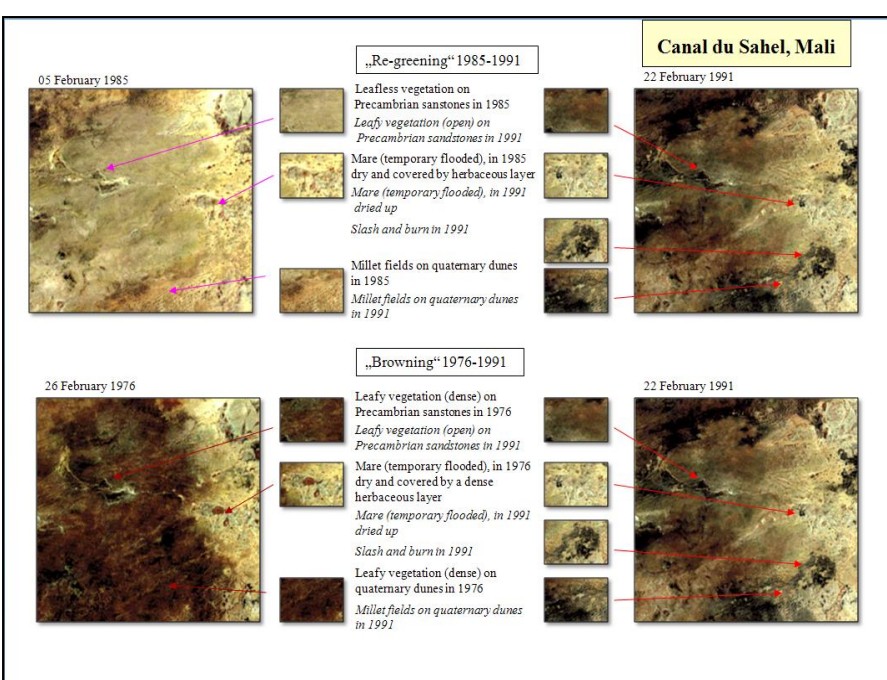

**Figure 12.** ~~Same satellite subsets demonstrate a much denser savanna vegetation still in the 1970s, indicating a post drought browning.~~ "Re-greening" (period 1985-1991) and "Browning" (period 1976-1991) visible in satellite subsets of the Canal du Sahel region.

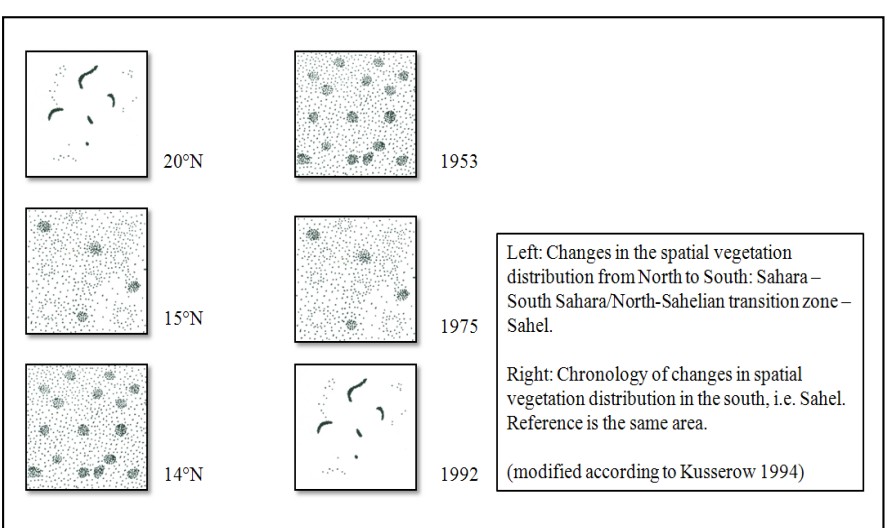

**Figure 13.** Left: Changes in the spatial vegetation distribution from North to South: Sahara – South Sahara/North-Sahelian transition zone – Sahel (modified according to Kusserow 1994). Right: Chronology of changes in spatial vegetation distribution in the south, i.e. Sahel (modified according to Kusserow 1994).

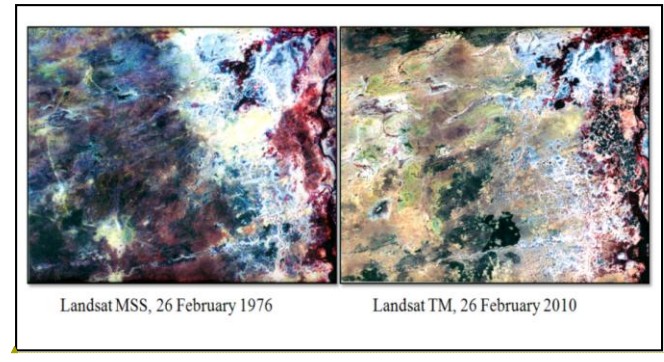

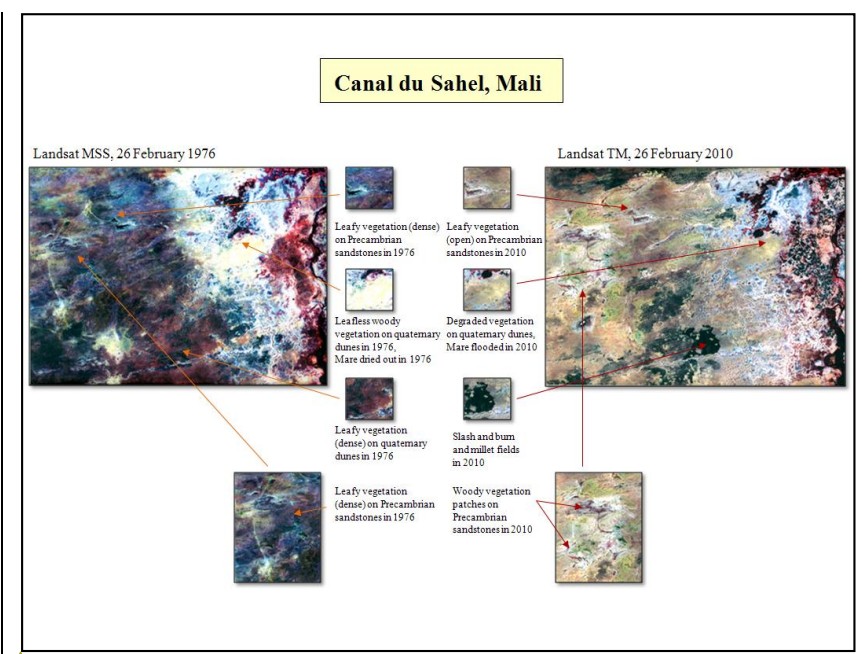

**Figure 14.** Development of vegetation pattern in the Canal du Sahel region in Mali over a period of 36 years (1976–2010, size of subset is approx. 80 km x 55 km).

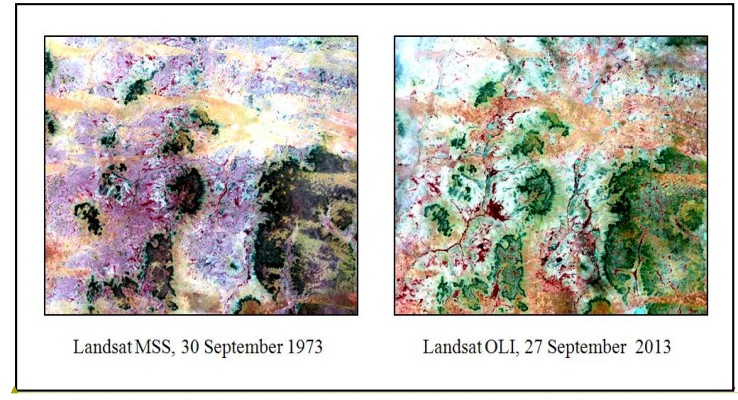

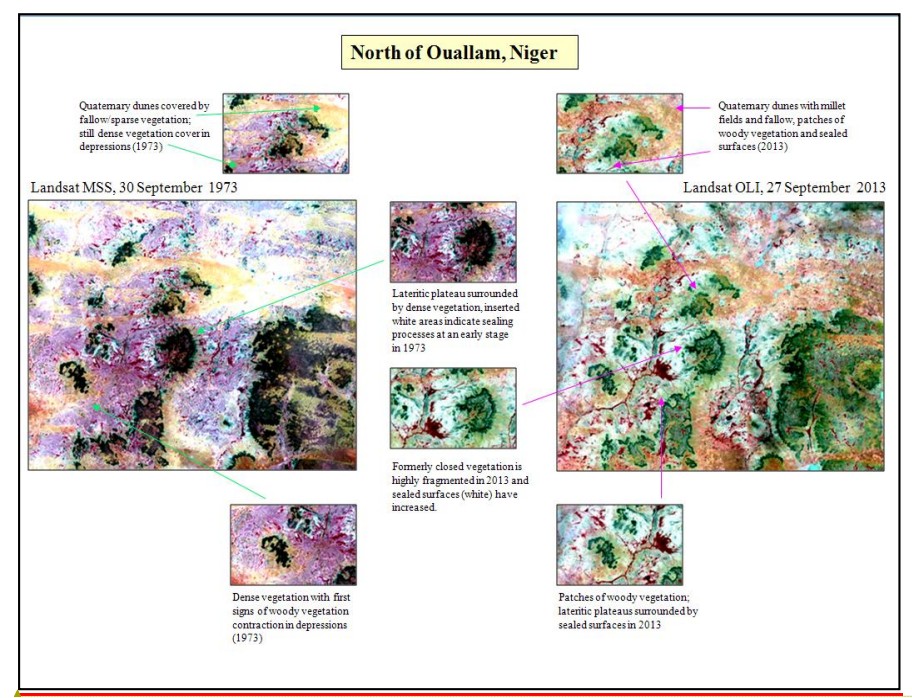

**Figure 15.** Comparison of two Landsat imageries of 1973 and 2013 showing how ligneous vegetation pattern formation has formed out of an originally uniform vegetation cover within 40 years (size of subset is approx. 60 km x 45 km).

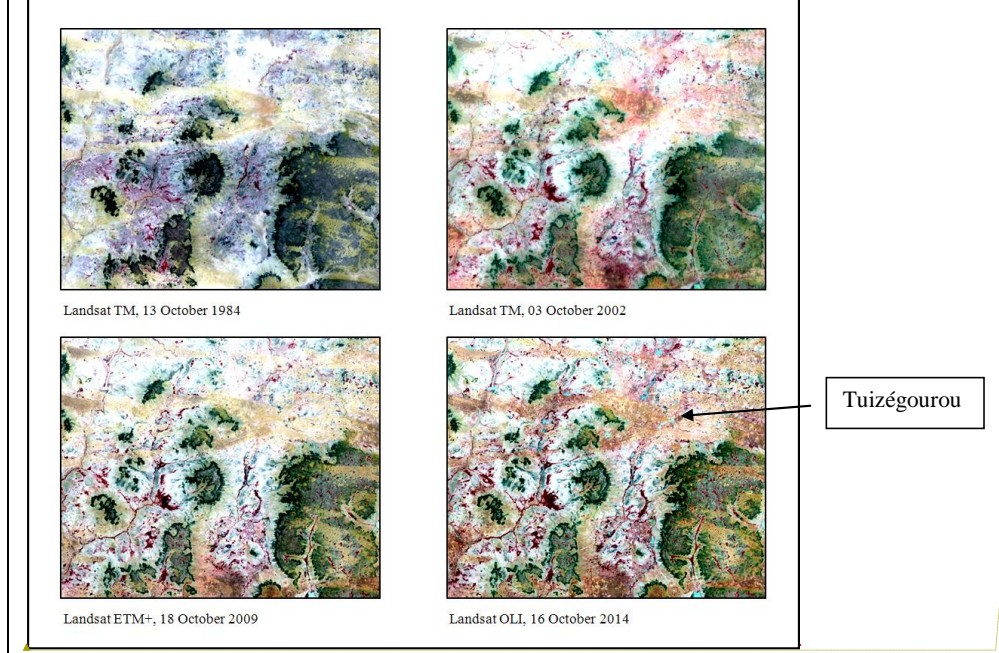

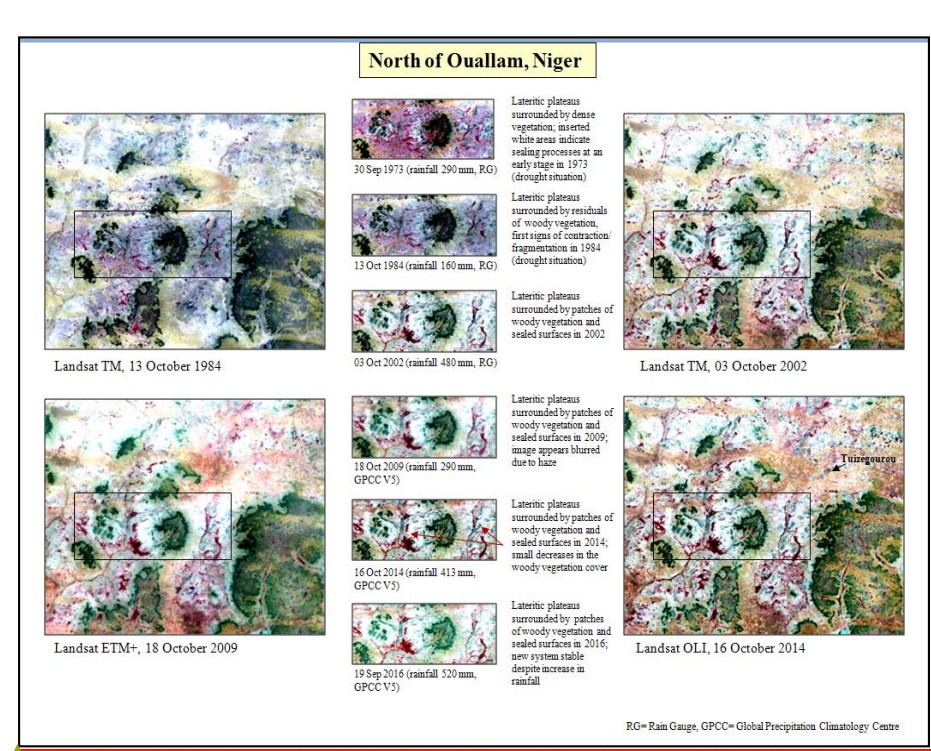

**Figure 16.** Pattern sequence of a research site in western Niger (North of Ouallam).

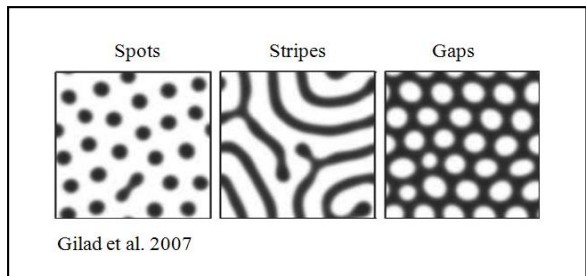

**Figure 17.** Model results show vegetation pattern development from an originally bare state into an uniformly distributed vegetation, reflecting system's ability of optimal self-organisation with respect to water resources (Gilad et al., 2007).

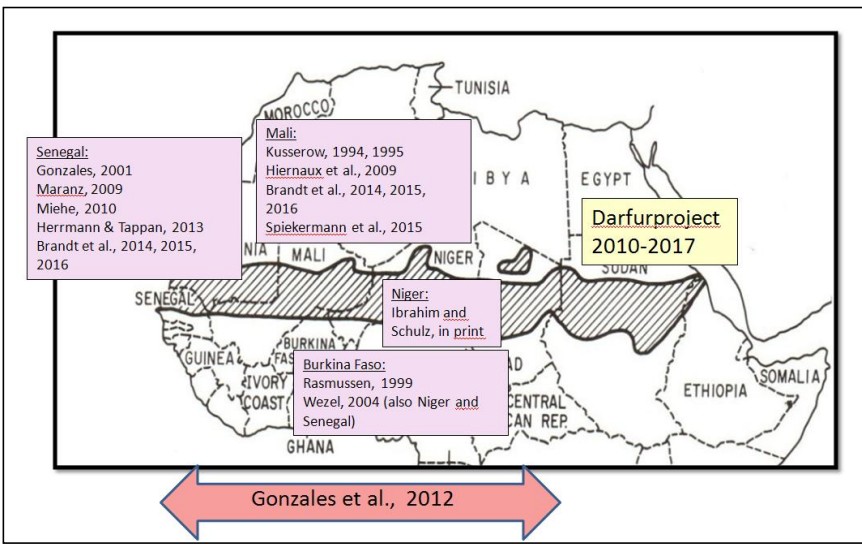

**Fig. 18:** Shift from pre-drought mesic to post-drought xeric plant species according to author and region.