# Peer review of "Desertification, Resilience and Re-greening in the African Sahel – A matter of the observation period?"

_Earth System Dynamics, 2017_

## Referee Comment (RC1) · M. Shoshany (Referee) · 17 May 2017

This paper presents an assessment of the Sahel Desertification debate using diverse information sources: historical documents, climatic records, satellite imagery and field studies. The author suggests rightly that in assessing desertification it is necessary to relate to pre-draught conditions. The re-greening hypothesis is assessed with regard to exiting studies (published articles) using evidence gathered from the sources listed earlier. The approach is objective and seems as to aim at gaining better understanding of the changes rather than approving a certain hypothesis. The article is most detailed and informative. The literature reviewed is excessive: I think that this is the most comprehensive synthesis on the topic of the Sahel desertification debate. Most

diverse anthropogenic and natural sources of desertification threats are discussed with reference to land use, land cover and ecological implications. The conclusions provide novel integration of known partial evidences:

- Recovery dependence on soil types is clearly a fundamental explanation. For sand soils the author suggests re-greening. In a way this conclusion leads to the question concern soil degradation as the main element and indicator of desertification.

- The decline of woody species biodiversity is presented as a primary element of desertification trend with turnover into more draught tolerant species.

- NDVI based evidence : the author own analysis suggest that if the data used is prior to 1980 there is a trend of declining NDVI rather than increasing NDVI.

- Woody species pattern changes due to fragmentation resulted increase in the distribution and size of bare surfaces.

I recommend publication, but suggest improvements regarding the figures presented: Figures representing comparison between satellite images acquired at different dates are important element of this study. However they must be based on comparable color scheme and calibration. Figure 8 is an excellent figure, but I recommend to position the information sources in a relative vertical position according to their type of indication: regreening above the main temporal axis and desertification below this axis.

---

## Referee Comment (RC2) · Anonymous Referee #2 · 10 Jun 2017

The paper covers the interaction between desertification, resilience and re-greening in the African Sahel, which is a subject relevant to Earth System Dynamics and the special issue on "Climate, land use, and conflict in Africa". It collects a vast range of information and data, drawn from a large number of publications as well as own field research. While this collective body of research is a valuable and solid contribution that offers interesting individual insights in this field, the way of presentation could be improved.

Overall the paper is very long, with around 18,000 words, including nearly 190 references which looks like a comprehensive literature review. Besides the introduction and

conclusion sections the paper puts everything else into the very long Section 2 and its five subsections, rather than making each of these a separate section.

While the length of the paper is not necessarily a critical issue, the presentation of methodologies and results appears rather unstructured and sometimes repetitive, adding numerous details about various vegetation patterns at different periods of time and locations in the Sahel, diverse sources of data (historical documents and records; various satellite images and resolutions; field observations from several authors) and other methods such as models of pattern formation. Jumping between the different combinations of these dimensions makes it hard to read, difficult to follow a logical track and recognize the overall picture among the many puzzle pieces.

Much of the paper refers to published work, looking like a comprehensive literature review, with some results based on own previous work scattered across the text. While this collection is useful it is not easy to identify own new results among results from others. In addition, it would be good to understand the role of modeling in this paper and to which degree it has been applied by the author.

To address these concerns and strengthen the paper, a more structured and perhaps condensed version would be a gain, making it more accessible to the reader and the synthesis clearer, to better see the overall picture. A systematical presentation of information would be helpful, e.g. in a table, that summarizes the results across the different dimensions (time, location, pattern/change, source/method), and/or with an integrated map highlighting where and when vegetation has changed in which way.

Besides more clearly stating the key messages, it is recommended to clarify what is new in this paper and what the own original contribution is in the context of the existing literature.

Some minor points: Check for more recent relevant references, especially on the Sahel greening. Check for language corrections, in particular in the abstract and in the beginning.

---

## Author Response (AR1)

**First reply to referee 1 (17-05-17)**

Dear Maxim, many thanks for reviewing my article and for your important recommendations.

I'll include more detailed information regarding color scheme and calibration into the text.

I have one question regarding figure 8, which in my text presents a compilation of selected botanical assessments in the West- and the East-African Sahel since 1900. I suggest that you mean another figure, maybe fig. 17? If so, I do not understand your question completely, please could you give me more details?

Kind regards Hannelore

**Second Reply to referee 1 (03-07-17)**

Regarding your comment "comparison of satellite imageries and processing":

Two different data sets were used in this article.The first one comprises historical datasets of Landsat MSS and TM (Example Mali, dates: 1976, 1985 and 1991). The raw data were bought from US Geological survey in 1985 and 1991 and were processed in 1991 using ERDAS 7.4.1 and 7.5 (Kusserow, 1990, 1994,1995). RGB = 4-2-1 (MSS) and 4-3-2 (TM).

Processing steps:
- Correction of six line effect for the MSS data.
- Master scene from 1991 was relatively corrected (haze correction).
- Geometric correction was conducted on the basis of the topographic map (UTM 31/WGS84) by using 17 way points (scene subset: 70 km x 50 km).
- MSS scenes was geometrically corrected on the basis of the master scene (TM).
- Relative calibration of the three datasets (radiometric correction) was performed by look up table modification (calibration on the basis of two test sites, showing no temporal variation).

Note: vegetation classification was done using visual interpretation techniques based on in situ knowledge.

The second data set includes Landsat MSS, TM, ETM+ and OLI data and was ordered for free from the United States Geological Survey through Earth Explorer (http://earthexplorer.usgs.gov/). These data are already pre-processed and systematically corrected. So, no further processing steps were necessary. The data sets comprise:

- Landsat MSS (example Burkina Faso and Niger: year 1973; RGB = 4-2-1).
- Landsat 4-5 Thematic Mapper (TM, example Niger: year 1984, 2002; RGB =4-3-2).
- Landsat (ETM+, example Niger: year 2009: RGB =4-3-2).
- OLI (example Niger: year 2013, 2014, example Burkina F.: year 2013; RGB =5-4-3).

The data were displayed with ENVI 4.7 (UTM 31/WGS84) using the default of a 2% linear stretch applied to each image band and for all data.

I still have a question regarding figure 8, which in my text presents a compilation of selected botanical assessments in the West- and the East-African Sahel since 1900. I suggest that you mean another figure, maybe fig. 17? If so, I do not understand your question completely, please could you give me more details?

Kind regards Hannelore

**First reply to referee 2 (19-06-17)**

Dear Referee,

thank you very much for reviewing my paper and your helpful comments.
To my understanding your comments refer to:
- (i)    a more clearer structure accentuation key messages

(ii)    underline new aspects

(iii)    clearer indicate what are the own contributions and results in the paper

(iv)    more condensed version

Further recommendations include more recent and relevant references and some language corrections (abstract and beginning).

Regarding your comments, I will work on the following:

(i)    key messages/synthesis will be clearer and better structured (by using table/fig.)

(ii)    new aspects in the paper will be better presented

(iii)    own contributions will be presented more clearly

(iv)    I belief that the historical aspect is an important part in the paper, however I will try to work on a more condensed version.

New relevant references will be included. English language will be rechecked (abstract and beginning).

Kind regards

Hannelore Kusserow

**Second reply to referee 2 (11-07-17)**

Dear Referee,

Thanks for having reviewed my article. Based on your comments I made the following changes in the paper:

(v)    The numbering of the sections was changed from Sect.1, Sect.2., 2.2, 2.3 etc. to Sect. 1 to Sect.7 (each of the former Subsections became a Section number)..

(vi)    For the new sections 3 to 6, it was made clearer what own research contributions are and what those of other authors. Now, each section starts with a description of own contributions followed by an analysis of results of other authors. Both contributions are marked (headlines). In addition, repetitive sentences were eliminated and a more condensed version was established.

(vii)    Section 7 was revised and now summarizes own results/key messages and emphasize new aspects. A new Figure 18 was added, showing vegetation changes according to author and region).

(viii)    New relevant references were included and English language was rechecked (Abstract and Introduction were partly rephrased).

Kind regards

Hannelore Kusserow

[revised manuscript text omitted]

Thornbush-Savanna with *Balanites aegyptiaca*, *Adansonia digitata*

Tiger bush with *Commiphora africana*, *Boscia senegalensis*, *Guiera senegalensis*, *Pterocarpus lucens*

Millet fields on fixed dunes with *Faidherbia albida*

Grass vegetation on fixed dunes with *Aristida sp.*, *Combretum glutinosum*

**Figure 6.** Vegetation and Landuse in the north-east of Upper Volta and lower Gourma, Mali, status 1976–1977, modified after Krings (1980).

[Figure]

**Figure 7.** Position of late Quaternary dune systems in the Sahel according to Mensching (1990).

[Figure]

**Figure 8.** Compilation of selected botanical assessments in the West- and the East-African Sahel since 1900.

[Figure]

**Figure 9.** Location of average isohyets at CLINO 1 (1931–1960) and CLINO 2 (1961–1990), position of research areas is indicated by the red box (modified after Kusserow and Oestreich, 1998).

[Figure]

**Figure 10.** Residuals of initially dense *Dalberghia melanoxylon* communities.

[Figure]

**Figure 11.** Satellite imageries from the Canal du Sahel region in Mali show the situation during the drought period (February 1985, see also view from the ground, June 1985) and during the so-called re-greening (February 1991, size of subset is approx. 70 km x 50 km).

[Figure]

**Figure 12.** Same satellite subsets demonstrate a much denser savanna vegetation still in the 1970s, indicating a post-drought browning.

[Figure]

**Figure 13.** Left: Changes in the spatial vegetation distribution from North to South: Sahara – South Sahara/North-Sahelian transition zone – Sahel (modified according to Kusserow 1994). Right: Chronology of changes in spatial vegetation distribution in the south, i.e. Sahel (modified according to Kusserow 1994).

[Figure]

**Figure 14.** Development of vegetation pattern in the Canal du Sahel region in Mali over a period of 36 years (1976–2010, size of subset is approx. 80 km x 55 km).

[Figure]

**Figure 15.** Comparison of two Landsat imageries of 1973 and 2013 showing how ligneous vegetation pattern formation has formed out of an originally uniform vegetation cover within 40 years (size of subset is approx. 60 km x 45 km).

[Figure]

**Figure 16.** Pattern sequence of a research site in western Niger (North of Ouallam).

[Figure]

**Figure 17.** Model results show vegetation pattern development from an originally bare state into an uniformly distributed vegetation, reflecting system's ability of optimal self-organisation with respect to water resources (Gilad et al., 2007).

[Figure]

**Fig. 18:** Shift from pre-drought mesic to post-drought xeric plant species according to  author and region.

[Figure]

[Figure]

**Figure 14.** Development of vegetation pattern in the Canal du Sahel region in Mali over a period of 36 years (1976–2010, size of subset is approx. 80 km x 55 km).

Landsat MSS, 30 September 1973    Landsat OLI, 27 September 2013

**Figure 15.** Comparison of two Landsat imageries of 1973 and 2013 showing how ligneous vegetation pattern formation has formed out of an originally uniform vegetation cover within 40 years (size of subset is approx. 60 km x 45 km).

Landsat TM, 13 October 1984    Landsat TM, 03 October 2002

Landsat ETM+, 18 October 2009    Landsat OLI, 16 October 2014

Tuizégourou

**Figure 16.** Pattern sequence of a research site in western Niger (North of Ouallam).

[Figure]

Left: Changes in the spatial vegetation distribution from North to South: Sahara – South Sahara/North-Sahelian transition zone – Sahel.

Right: Chronology of changes in spatial vegetation distribution in the south, i.e. Sahel. Reference is the same area.

(modified according to Kusserow 1994)

**Figure 17.** Left: Changes in the spatial vegetation distribution from North to South: Sahara – South Sahara/North-Sahelian-transition zone – Sahel (modified according to Kusserow 1994). Right: Chronology of changes in spatial vegetation distribution in the south, i.e. Sahel (modified according to Kusserow 1994).

---

## Author Response (AR2)

Author's Response (H. Kusserow)

**Editor Decision: Publish subject to minor revisions (review by Editor)** (04 Sep 2017) by Tim Brücher
Comments to the Author:
Dear Author,
thanks for your updated version. As one reviewer asked to put figures on the same color scheme for better comparison I would like to ask you to (i) change the color if possible, or (ii) put a color bar next each figure, so it is easy to see, which color has which meaning - otherwise it is to difficult to compare the images.
A color bar should be attached to all of these figures - although the meaning is explained in the text.
(i) Color bars/legend were added to Fig. 4, 12, 14-16. Fig. 12 has been redesigned.

Second: The Figure 7 was asked to be on a vertical structure:
This would enable to have East Africa on the right hand side, West Africa on the left hand side and time running from top to bottom. So it is much more appropriate to put a tick mark on the time axis to the corresponding entry on the Literature.

(ii) Fig. 8 (not fig. 7) has been completely redesigned according to the editors' recommendation.

For the structure of the manuscript I would recommend a complete data+method section after introduction, starting with existing data and new (own) data as well as explaining all methods used / developed in this article. This should not be included in the Introduction Section.
It would also help to put the Result-section (discussion) in one chapter with subchapters and close with a Final Conclusion Chapter:

1) Introduction
2) Data & Method
3.1 ... 3.5) Result-Section (If necessary for grouping, you can have 2 chapters (3+4) for the sesult-section.
last section: Key-Messages ...

(iii) A complete data+method chapter has been inserted as Sect. 2. The structure of the paper is now:

1) Introduction
2) Data and Methods
3) Desertification, Resilience and Regreening – a general overview
4) Desertification, Resilience and Regreening – key factors
   4.1 The relevance of edaphic factors for a vegetation re-greening
   4.2 Plant species change
   4.3 Re-greening Sahel based on NOAA-AVHRR, GIMMS3g, MODIS, SPOT VGT studies
   4.4 Changes in vegetation pattern, self-organised patchiness
5) Key messages and new aspects

(iv) Listing of changes done in the document:

- Color bars/legend were added to Fig. 4, 12, 14-16, Fig. 12 has been redesigned
- Fig. 8 was completely redesigned, add-on of further authors
- Correction of section numbering
- Information on data and methods in the main text was transferred into Section 2
- Corrections of spelling mistakes in the main text
- Revision of references according to Copernicus Publications Reference Types and add-on of 9 further authors